# Homocysteine directly interacts and activates the angiotensin II type I receptor to aggravate vascular injury

Tuoyi Li[1,2], Bing Yu[1], Zhixin Liu[3], Jingyuan Li[4], Mingliang Ma[3], Yingbao Wang[1], Mingjiang Zhu[5], Huiyong Yin [5], Xiaofeng Wang[4], Yi Fu[1], Fang Yu[1], Xian Wang[1], Xiaohong Fang[6], Jinpeng Sun[3,7] & Wei Kong[1]

Hyperhomocysteinemia (HHcy) is a risk factor for various cardiovascular diseases. However, the mechanism underlying HHcy-aggravated vascular injury remains unclear. Here we show that the aggravation of abdominal aortic aneurysm by HHcy is abolished in mice with genetic deletion of the angiotensin II type 1 (AT1) receptor and in mice treated with an AT1 blocker. We find that homocysteine directly activates AT1 receptor signalling. Homocysteine displaces angiotensin II and limits its binding to AT1 receptor. Bioluminescence resonance energy transfer analysis reveals distinct conformational changes of AT1 receptor upon binding to angiotensin II and homocysteine. Molecular dynamics and site-directed mutagenesis experiments suggest that homocysteine regulates the conformation of the AT1 receptor both orthosterically and allosterically by forming a salt bridge and a disulfide bond with its Arg[167] and Cys[289] residues, respectively. Together, these findings suggest that strategies aimed at blocking the AT1 receptor may mitigate HHcy-associated aneurysmal vascular injuries.

[1] Department of Physiology and Pathophysiology, School of Basic Medical Sciences, Peking University; Key Laboratory of Molecular Cardiovascular Science, Ministry of Education, Beijing 100191, China. [2] Capital Normal University High School, Beijing 100048, China. [3] Department of Biochemistry and Molecular Biology, School of Medicine, Shandong University; Key Laboratory Experimental Teratology of the Ministry of Education, Jinan, Shandong 250012, China. [4] CAS Key Laboratory for Biological Effects of Nanomaterials and Nanosafety, Institute of High Energy Physics, 19 B, Yuquan Road, Beijing 100049, China. [5] Key Laboratory of Food Safety Research, Institute for Nutritional Sciences (INS), Shanghai Institutes for Biological Sciences (SIBS), Chinese Academy of Sciences (CAS), Shanghai 200031, China. [6] Key Laboratory of Molecular Nanostructure and Nanotechnology, CAS Research/Education Center for Excellence in Molecular Sciences, Institute of Chemistry, Chinese Academy of Sciences, Beijing 100190, China. [7] School of Medicine, Duke University Medical Center, Durham, NC 27710, USA. Tuoyi Li, Bing Yu, and Zhixin Liu contributed equally to this work. Correspondence and requests for materials should be addressed to J.S. (email: sunjinpeng@sdu.edu.cn) or to W.K. (email: kongw@bjmu.edu.cn)

Homocysteine (Hcy) is a sulfur-containing, non-essential amino acid derived from the essential amino acid methionine and is actively involved in numerous biochemical reactions. Hyperhomocysteinemia (HHcy, circulating Hcy ≥15 μM) is an established independent risk factor for a variety of vascular diseases, including myocardial infarction, stroke, and abdominal aortic aneurysm (AAA), among others[1–3]. AAA is one of the leading causes of sudden death in aging men and lacks any proven drug therapy. Using a mouse model, we recently reported that HHcy significantly aggravated angiotensin II (Ang II)-induced and $CaPO_4$-evoked AAA, while folic acid supplementation ameliorated these effects[4,5]. Although compelling evidence has indicated that HHcy initiates vascular inflammation, damages endothelial cells, promotes medial proliferation, facilitates adventitial activation, and disturbs hemostasis/coagulation[4,6,7], the mechanism underlying the aggravation of vascular injury by HHcy remains elusive.

The renin-angiotensin-aldosterone system (RAAS) plays an essential role in vascular pathogenesis. Ang II, the primary mediator of the RAAS, exerts its diverse bioactive effects primarily by activating the AT1 receptor (Ang II type 1 receptor), a G-protein-coupled receptor. Genetic deletion of $AT1a$ receptor effectively prevents pathological vascular injuries in a variety of animal models, including models of atherosclerosis, hypertension, and AAA[8–10]. Accordingly, antagonism of the AT1 receptor by drugs of the sartan family is extensively used for the prevention or treatment of cardiovascular diseases. Of interest, in addition to Ang II, factors, such as mechanical stretch, interaction with autoantibodies, or artificial substitution of certain amino acids (e.g., $Asn^{111}$) of the AT1 receptor are known to constitutively activate the AT1 receptor and increase downstream signaling even in the absence of Ang II[11]. Additionally, allosteric modulation, which is defined as a ligand binding to a site different from its endogenous ligand binding site and exerting positive or negative effects on the affinity or efficacy of the natural ligand, was also found to participate in AT1 receptor regulation[12]. However, under pathological conditions, particularly in HHcy, whether Hcy at pathological concentrations directly activates the AT1 receptor or allosterically regulates the AT1 receptor and subsequently contributes to vascular injuries are unknown. Previous studies have shown that Hcy upregulated the transcription of the AT1 receptor and other RAAS components[13,14]. Here we identified a novel regulatory mechanism that homocysteine directly interacts and activates the angiotensin II type I receptor to aggravate vascular injury.

## Results

### The AT1a receptor mediates HHcy-aggravated vascular injury.
We previously reported that HHcy aggravated Ang II infusion-or periadventitial $CaPO_4$-induced vascular inflammation and AAA formation in mice, respectively[4,5]. Here we investigated whether the AT1 receptor mediates HHcy-aggravated vascular injury in vivo in two different models: elastase-induced and $CaPO_4$-induced AAA mouse models.

In the elastase model, 8-week-old male wild type (WT) and $AT1a^{-/-}$ mice were given Hcy (1.8 g/L) in drinking water for a total of 28 days. Fourteen days after Hcy application, the mice underwent surgery to induce AAA, and the aortas were collected 14 days after surgery (Supplementary Fig. 1a). Hcy supplementation resulted in mild to moderate HHcy in both WT and $AT1a^{-/-}$ mice (plasma total Hcy: WT HHcy vs. WT CTL: $25.28 \pm 2.13$ ($n = 12$) vs. $8.44 \pm 0.75$ ($n = 12$) μM; $AT1a^{-/-}$ HHcy vs. $AT1a^{-/-}$ CTL: $23.60 \pm 2.70$ ($n = 9$) vs. $8.10 \pm 0.50$ ($n = 12$) μM), without significant changes in body weight, blood pressure, plasma Ang II concentration, or total cholesterol and triglyceride levels

(Supplementary Table 1). As expected, elastase induced the expand of abdominal aorta (WT vs. WT + elastase: $0.89 \pm 0.04$ ($n = 5$) vs. $1.28 \pm 0.01$ ($n = 12$) mm) and the degradation of aortic elastin in WT mice, whereas elastase-induced abdominal aorta enlargement was inhibited in AT1a deficient mice ($AT1a^{-/-}$ vs. $AT1a^{-/-}$ + elastase: $0.88 \pm 0.05$ ($n = 5$) vs. $0.99 \pm 0.05$ ($n = 12$), no significance). Furthermore, HHcy markedly enhanced the enlarged maximal abdominal aortic diameters and aggravated elastin degradation of WT mice, whereas, $AT1a$ receptor knockout significantly ameliorated HHcy-aggravated vascular injury and aneurysm formation induced by elastase (Fig. 1a–d).

To further verify the above observations, $CaPO_4$ surgery was performed on WT mice and $AT1a^{-/-}$ mice to induce AAA following Hcy or water supplementation (Supplementary Fig. 1a). Consistently, mild to moderate HHcy was induced in both WT and $AT1a^{-/-}$ mice as evidenced by the elevated levels of plasma total Hcy (Supplementary Table 2). The $CaPO_4$-induced AAA was aggravated by HHcy in WT mice but not in $AT1a^{-/-}$ mice (Supplementary Fig. 2a–c). Moreover, application of the AT1 receptor blocker telmisartan (10 mg/kg/d in drinking water for 28 days, Supplementary Fig. 1b, Supplementary Table 3), abolished HHcy-enhanced AAA formation in WT mice (Supplementary Fig. 3a–c), reinforcing that the AT1 receptor mediated HHcy-aggravated vascular aneurysmal injury.

A previous study showed that the vessel wall resident AT1a receptor predominantly contributes to AAA formation in mice[15]. Moreover, proinflammatory MCP-1/IL-6 and matrix metalloproteinase 9 (MMP-9) were upregulated by HHcy and aggravated aneurysmal vascular inflammation[4]. In addition, genetic deletion of the MCP-1 receptor C–C motif chemokine receptor 2 (CCR2), IL-6, or MMP-9 ameliorated aortic aneurysm pathogenesis[16,17]. Therefore, using ex vivo aortic ring explant cultures, we examined MCP-1, IL-6 secretion and MMP-2, MMP-9 activation to address how HHcy regulated the vascular AT1 receptor. Similar as Ang II (1 μM), Hcy at 100 μM markedly induced MCP-1 and IL-6 secretion from the aortic ring explants, starting as early as 30 min and lasting for 12 h (Supplementary Fig. 4a–d). However, these effects were not observed in the aortic ring explants from $AT1a^{-/-}$ mice (Fig. 1e, f). Similarly, gelatin zymography showed that MMP-2 and MMP-9 were activated by Hcy in WT mice aortic rings, but this activation was inhibited in $AT1a^{-/-}$ mice aortic rings (Fig. 1g, h). Notably, an mRNA analysis of mouse aortic rings after Hcy treatment from 6 to 48 h revealed no alteration in RAAS components within 12 h, including AT1a, angiotensinogen (AGT), angiotensin-converting enzyme (ACE) and renin, although their expression was upregulated after a longer stimulation time (24 or 48 h) (Supplementary Fig. 5a–d). Accordingly, Ang II secretion of aortic ring explants was not altered by Hcy within 24 h but significantly elevated under 48-h treatment of Hcy (Supplementary Fig. 5e). Together, these results suggest that Hcy regulates the vascular AT1 receptor through at least two distinct mechanisms: transcription-dependent pathways for a long period of stimulation and transcription-independent pathways for short-term Hcy treatment.

### Hcy activates AT1 receptor signaling.
We next asked whether Hcy directly activated the AT1 receptor upon a short period of stimulation. In addition to the natural ligand of the AT1 receptor, Ang II, several factors induce constitutive AT1 receptor signaling in the absence of Ang II, such as the membrane environment, interacting proteins, receptor autoantibodies, and single-nucleotide polymorphisms[11]. In general, two major downstream signaling pathways, G-protein signaling and arrestin-mediated pathways, are responsible for the functions of the AT1 receptor[18–21]. To determine whether Hcy could directly activate

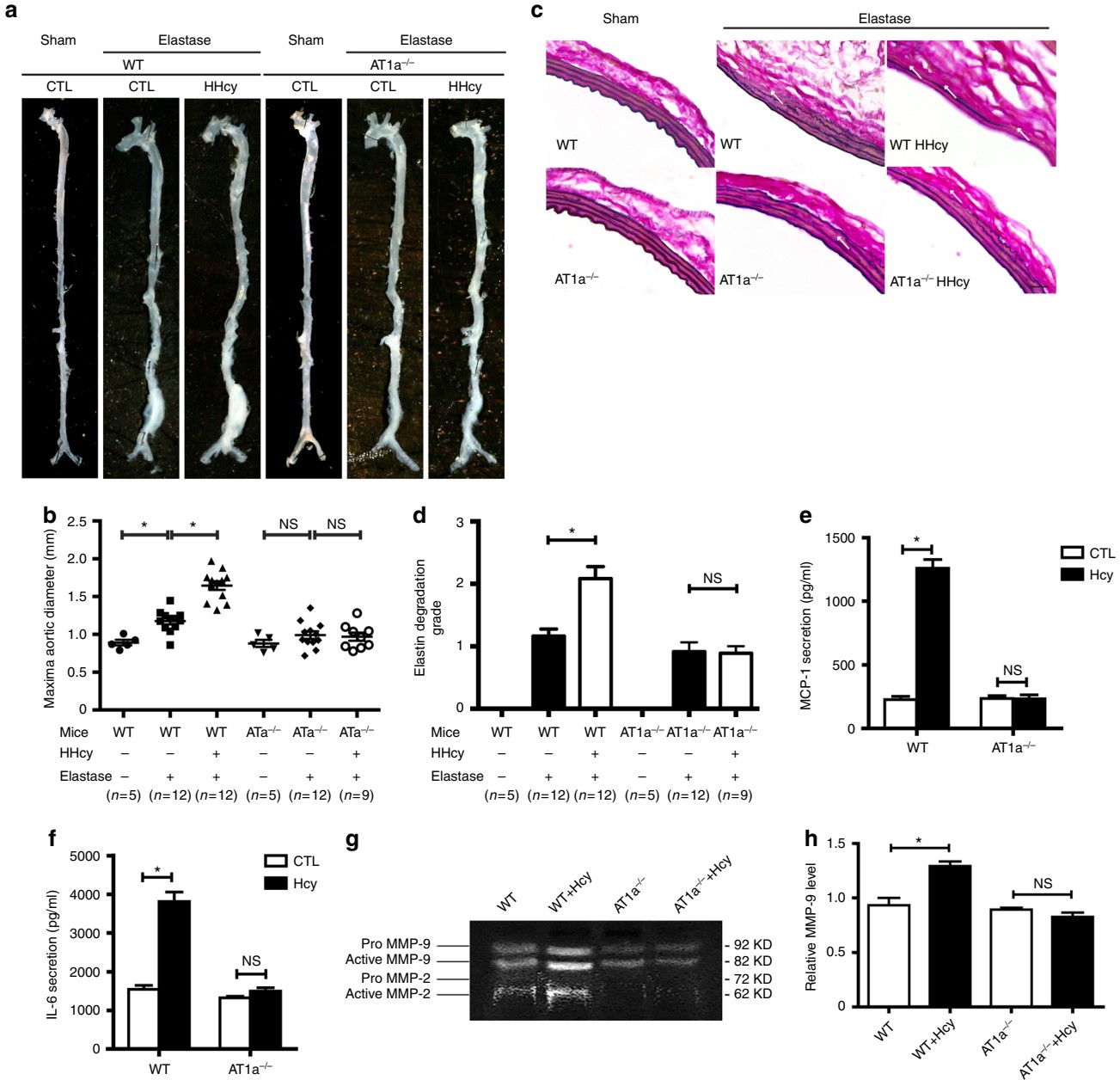

**Fig. 1** HHcy aggravates AAA in mice and induces vascular injury through the AT1a receptor. **a** Representative photographs of elastase-induced AAA in WT mice and $AT1a^{-/-}$ mice with or without elastase treatment. HHcy, Hcy (1.8 g/L) in drinking water plus elastase treatment. $N = 5$–12. **b** The quantification of the infrarenal abdominal aortic diameters in mice with elastase-induced AAA. The data represent as mean ± SEM. $N = 5$–12, *$P < 0.05$, Kruskal–Wallis test followed by Dunn's test. **c** Representative Verhoeff–Von Gieson staining of elastin degradation. Scale bar, 20 μm. $N = 5$–12. **d** The quantification of elastin degradation. The data represent as mean ± SEM. $N = 5$–12, *$P < 0.05$, Kruskal–Wallis test followed by Dunn's test. **e**, **f** Hcy (100 μM)-induced WT and $AT1a^{-/-}$ mice abdominal aortic ring MCP-1 secretion and IL-6 secretion 60 min after stimulation. The data represent as mean ± SEM. $N = 9$–12. *$P < 0.05$, two-way ANOVA followed by the Bonferroni post hoc test. **g**, **h** Representative gelatin zymography and quantification (active MMP-9) of conditioned medium of abdominal aortic rings from WT and $AT1a^{-/-}$ mice after Hcy (100 μM) ex vivo stimulation for 20 h. The data represent as mean ± SEM. $N = 6$. *$P < 0.05$, two-way ANOVA followed by the Bonferroni post hoc test

the AT1 receptor, we assessed G-protein-dependent signaling, including PKC and ERK1/2 phosphorylation, intracellular $Ca^{2+}$ levels and NFAT signaling, as well as arrestin-mediated activation, including β-arrestin 2 translocation and receptor internalization, in HEK293A cells transfected with the human AT1 receptor. Compared to Ang II (Supplementary Fig. 6a–c), Hcy at 100 μM resulted in delayed but sustained PKC and ERK1/2-MAPK phosphorylation, which was abolished by pretreatment with the AT1 receptor blocker telmisartan (1 μM) (Fig. 2a–e). Similar to Ang II (Supplementary Fig. 6d), Hcy dose-dependently

activated intracellular $Ca^{2+}$ signaling, which was blocked by telmisartan pretreatment (Fig. 2f, Supplementary Fig. 7a, Supplementary Table 4). Dual-luciferase reporter assays of NFAT were then conducted to demonstrate that Hcy dose-dependently activated intracellular NFAT signaling (Fig. 2g, Supplementary Table 5), although to a lesser extent than Ang II (Supplementary Fig. 7b). Telmisartan pretreatment reversed Hcy-induced NFAT activation (Fig. 2g, Supplementary Table 5). In parallel, similar to Ang II (Supplementary Fig. 8a, b), Hcy stimulation for 15 min caused β-arrestin 2 translocation, co-localization with the AT1

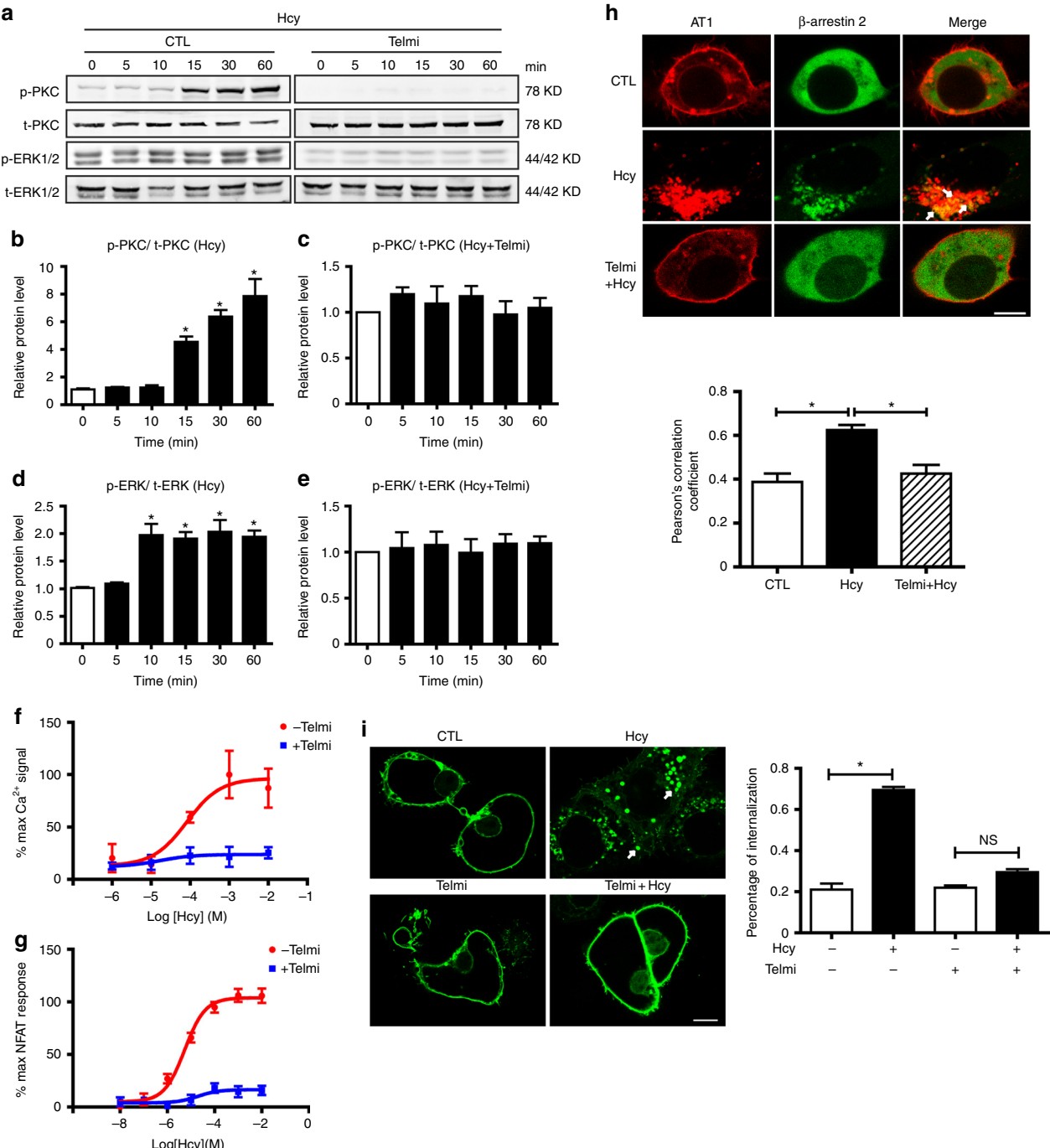

**Fig. 2** Hcy activates the AT1 receptor downstream $G_q$ and β-arrestin 2 signaling pathways. **a–e** Representative western blots and quantification of phosphorylated and total PKC and ERK1/2 in Hcy (100 μM)-treated HEK293A cells (transfected with the human AT1 receptor) with or without telmisartan (1 μM) pretreatment. The data represent as mean ± SEM, $N = 6$. *$P < 0.05$, one-way ANOVA followed by the Bonferroni post hoc test. **f** $Ca^{2+}$ signaling in HEK293A cells (transfected with the human AT1 receptor) stimulated by Hcy with or without telmisartan (1 μM) pretreatment, detected using Fluo 3-AM and quantified by the Leica confocal microscope system. The data represent as mean ± SEM, $N = 6$. **g** Hcy-induced NFAT signaling with or without telmisartan (1 μM) pretreatment, detected by the dual-luciferase assay system (Promega). The data represent as mean ± SEM, $N = 6$. **h** Representative fluorescence and quantification of Hcy (100 μM)-induced co-localization of the AT1 receptor (overexpression of human AT1-mCherry plasmid; red) and β-arrestin 2 (overexpression of β-arrestin 2-GFP plasmid; green) in cultured HEK293A cells. The merged area is indicated as yellow (arrows). Scale bar, 10 μm. The data represent as mean ± SEM, $N = 6$, *$P < 0.05$, one-way ANOVA followed by the Bonferroni post hoc test. **i** Representative fluorescence and quantification of Hcy (100 μM)-induced AT1 receptor (overexpression of mouse AT1a-GFP plasmid; green) internalization (arrows) with or without telmisartan pretreatment in COS7 cells. The data represent as mean ± SEM, $N = 10$, *$P < 0.05$, two-way ANOVA followed by the Bonferroni post hoc test

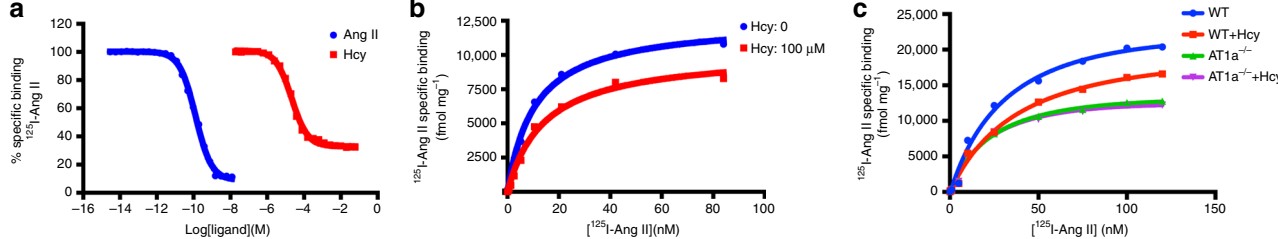

**Fig. 3** Hcy directly interacts with the AT1 receptor. **a** The specific binding of [$^{125}$I]-Ang II and the AT1 receptor with different competing concentrations of Ang II and Hcy. The data represent as mean ± SEM, $N = 6$. **b** The saturation binding curve of [$^{125}$I]-Ang II and the AT1 receptor in the presence and absence of Hcy (100 μM). The data represent as mean ± SEM, $N = 6$. **c** The saturation binding curve of [$^{125}$I]-Ang II and the AT1 receptor in membrane proteins extracted from WT and $AT1a^{-/-}$ aortas of mice fed with or without Hcy. The data represent as mean ± SEM, $N = 6$

receptor (Fig. 2h), as well as AT1 internalization (Fig. 2i), which was reversed by telmisartan pretreatment. In contrast, AT1 activation was not observed in cysteine-treated groups (Supplementary Fig. 9). As telmisartan also has PPARγ activation effect[22], we next applied PPARγ agonist rosiglitazone (RSG) to examine whether it inhibit the activation of AT1 receptor by Hcy. Distinct from telmisartan, RSG did not inhibit Hcy-induced PKC and ERK phosphorylation and NFAT activation, implying that the inhibitive effect of telmisartan was independent on its side-effect on activating PPARγ (Supplementary Fig. 10a, b). Furthermore, similar to telmisartan, other sartans including candesartan and losartan markedly blocked Hcy-induced PKC and ERK phosphorylation and NFAT activation (Supplementary Fig. 10c, d).

To exclude the possibility that Hcy-induced AT1 receptor activation could be an indirect effect following Ang II secretion, we measured the Ang II concentration in the conditioned media and used enalapril, an ACE inhibitor, to inhibit Ang II production. Hcy did not affect Ang II production in HEK293A cells between 5 and 60 min of treatment (Supplementary Fig. 11a). In particular, enalapril pretreatment did not affect Hcy-induced NFAT activation (Supplementary Fig. 11b) or PKC and ERK phosphorylation (Supplementary Fig. 11c–e). Therefore, Hcy activated the AT1 receptor independent of Ang II secretion. To further exclude the role of Ang II production involved in HHcy-aggravated AAA formation in vivo, we compared HHcy-exacerbated AAA formation with and without administration of enalapril in elastase-induced aneurysmal models (Supplementary Fig. 12a). As expected, the administration of enalapril (30 mg/kg per day) in drinking water significantly decreased the blood pressure in both control and HHcy mice, as well as profoundly inhibited Ang II production and enhanced the plasma renin activity (Supplementary Table 6). In contrast, administration of enalapril displayed no effect on HHcy-enhanced aneurysmal aortic dilation (Supplementary Fig. 12b, c), indicating that HHcy-aggravated AAA formation is independent on Ang II production.

Previous studies have indicated that HHcy induces reactive oxygen species (ROS) production in various cells, including endothelial cells, vascular smooth muscle cells and adventitial fibroblast cells[4,23,24]. We next asked whether Hcy-induced AT1 receptor activation through indirect ROS production. Therefore, the $H_2O_2$ scavenger catalase and the NADPH oxidase inhibitor DPI were applied. Both catalase and DPI treatment inhibited Hcy-induced $H_2O_2$ production, as measured by Amplex Red assays (Supplementary Fig. 13a). In contrast, neither catalase nor DPI showed any effect on Hcy-induced NFAT activation (Supplementary Fig. 13b) or PKC and ERK phosphorylation (Supplementary Fig. 13c–e). Thus, Hcy-induced AT1 receptor activation was not secondary to ROS production.

**Hcy directly interacts with the AT1 receptor**. We therefore hypothesized that Hcy directly interacted with the AT1 receptor.

Due to the difficulties in Hcy radiolabeling, including the lack of a phenolic hydroxyl group and its nature to be easily oxidized[25,26], we alternatively conducted radioligand competition binding assays and saturation binding assays in the presence or absence of Hcy with an extracted cellular membrane fraction transfected with the AT1 receptor. Both Ang II and Hcy were able to displace [$^{125}$I]-Ang II, yielding $pK_i$ values of 10.2 ± 0.020 and 4.97 ± 0.025, respectively (mean ± SEM, Fig. 3a). These data indicated that Hcy is a competitive ligand of Ang II and binds to AT1 receptor in its orthosteric site.

In addition to directly binding to the orthosteric site, we noticed that Hcy was not a typical competitive ligand of Ang II. Hcy did not replace radio-labeled Ang II as completely as unlabeled Ang II. Consistently, in the saturation binding assay, Hcy (100 μM) decreased the $B_{max}$ value but increased the $K_D$ value of [$^{125}$I]-Ang II ($B_{max}$ of Ang II vs. Hcy + Ang II: 12,519 ± 255.9 vs. 10,235 ± 362.3 fmol/mg protein, non-specific binding was subtracted; $K_D$ of Ang II vs. Hcy + Ang II: 10.76 ± 0.681 vs. 14.54 ± 1.473 nM, Fig. 3b). The Hcy-induced $B_{max}$ decrease at the saturated concentration of Ang II suggested that Hcy and Ang II were able to bind to the AT1 receptor simultaneously. Therefore, Hcy may also regulate Ang II-AT1 receptor signaling allosterically in addition to binding at the orthosteric site.

To further confirm the direct interaction of Hcy and AT1a receptor in vivo, we performed saturation binding assays with aortic membrane fraction of WT and $AT1a^{-/-}$ mice with or without HHcy. As results, HHcy attenuated the $B_{max}$ value but increased the $K_D$ value of [$^{125}$I]-Ang II in interaction with WT aortic membrane fraction ($B_{max}$ of control vs. Hcy: 26,268 ± 697 vs. 22,315 ± 618 fmol/mg protein, non-specific binding was subtracted; $K_D$ of control vs. Hcy: 36.4 ± 4.6 vs. 43.2 ± 3.9 nM. Fig. 3c). In contrast, whereas $AT1a$ knockout decreased both $B_{max}$ and $K_D$ values of [$^{125}$I]-Ang II binding with or without incubation with Hcy, it disrupted the effect of HHcy on $B_{max}$ and $K_D$ values of [$^{125}$I]-Ang II binding within aortic membrane fractions ($B_{max}$ of $AT1a^{-/-}$ vs. $AT1a^{-/-}$ +Hcy: 17,286 ± 526 vs. 16,962 ± 280 fmol/mg protein, non-specific binding was subtracted, no significance; $K_D$ of $AT1a^{-/-}$ vs. $AT1a^{-/-}$ +Hcy: 34.3 ± 4.9 vs. 36.2 ± 3.2 nM, no significance, Fig. 3c). The HHcy-induced $B_{max}$ attenuation at the saturated concentration of Ang II suggested that Hcy directly interact with the aortic AT1a receptor in vivo as well.

**Hcy modulates the conformation of AT1 receptor**. To provide further structural insight into Hcy-induced AT1 receptor activation, we generated a series of FlAsH BRET probes by incorporating the CCPGCC into the three AT1 receptor intracellular loops (ICLs). The CCPGCC paired with a luciferase domain at the AT1 receptor C-terminal end enabled us to monitor ligand-induced specific conformational changes of the AT1 receptor (Fig. 4a). Such intramolecular BRET has been successfully applied

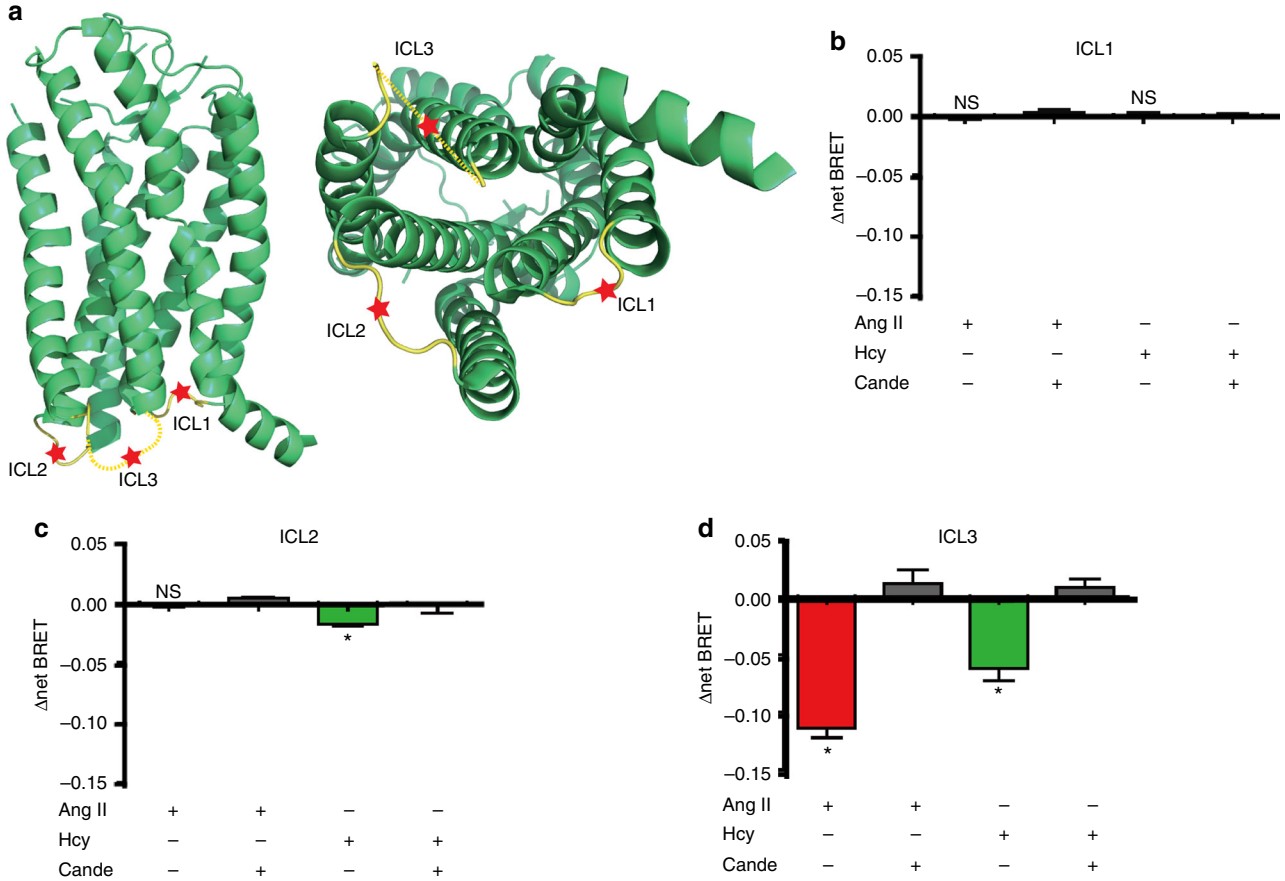

**Fig. 4** Hcy and Ang II induce different conformational changes of the AT1 receptor. **a** Schematic diagram of AT1 receptor-ICL-FlAsH-Rluc constructs. **b–d** Intramolecular $\Delta$net BRET measured in HEK293A cells (transfected separately with the three constructs ICL1–3) stimulated by Ang II (1 μM) or Hcy (100 μM) with or without candesartan (0.1 μM) pretreatment. The data represent as mean ± SEM, $N = 6$, *$P < 0.05$, one-way ANOVA followed by the Bonferroni post hoc test. ICL, intracellular loop

in the characterization of the conformational changes of GPCR signaling in previous studies[27–29]. Here, insertion of these FlAsH motifs did not significantly impair $G_q$ signaling or β-arrestin recruitment (data not shown), suggesting that these mutants did not destroy the structural integrity of the AT1 receptor. While Ang II did not induce a significant change in the BRET signal at ICL1 and ICL2, it significantly decreased the intramolecular BRET signal at ICL3, which may indicate the separation of the C terminus from ICL3 due to Ang II stimulation (Fig. 4b–d). Intriguingly, while Hcy-induced a decrease of the intramolecular BRET signal at ICL3 similar to Ang II, Hcy also significantly decreased the BRET signal at ICL2, suggesting that ICL2 moved away from the C terminus, which was specifically induced by Hcy but not Ang II (Fig. 4b–d). Importantly, these specific conformational changes induced by either Ang II or Hcy were totally eliminated by pre-incubation of the cells with the AT1 receptor antagonist candesartan (0.1 μM), confirming a direct correlation of the ligand binding to the receptor and the observed conformational changes. Taken together, these different Rluc-AT1 receptor-FlAsH BRET signatures demonstrated that Hcy and Ang II activate AT1 receptor signaling through distinct conformational alterations on the intracellular side of the AT1 receptor.

**Arg[167] of AT1 mediates Hcy-induced AT1 receptor activation.** To further reveal the molecular details of the interaction between Hcy and the AT1 receptor, we conducted molecular docking to predict the potential binding site of Hcy on the AT1 receptor

based on the crystal structure of the human AT1 receptor and its antagonist ZD7155 (PDB code: 4YAY)[30].

By global docking, 5 clusters of structures with a root mean square deviation (RMSD) <2 Å were generated (i.e., clusters 1–5, Fig. 5a i–v). Local docking was further conducted in cluster 3 (Fig. 5a iii), as Hcy in cluster 3 was sitting in the known binding pocket (for antagonist ZD7155) of the AT1 receptor[30] and the RMSD of both AT1 receptor and Hcy were the lowest in cluster 3 (Supplementary Fig. 14a, b). The final model indicated that the binding stability of Hcy could be predominantly attributed to a salt bridge formed with Arg[167] (Fig. 5b), which is a known binding site of the AT1 receptor with the phenolic group of tyrosine in Ang II[31]. Accordingly, we mutated Arg[167] by site-directed mutagenesis (Fig. 5c) and conducted competitive radioligand binding assays. As shown in Fig. 5d, the R167A mutation significantly inhibited Hcy and AT1 receptor binding (pKᵢ of Hcy-WT AT1 vs. Hcy-R167A AT1: 4.678 ± 0.024 vs. 4.036 ± 0.028, mean ± SEM, Fig. 5d). Accordingly, the AT1 R167A mutation completely blocked both Ang II- and Hcy-induced NFAT activation (Fig. 5e). Furthermore, BRET assay demonstrated that both Ang II and Hcy significantly induced β-arrestin 2 recruitment, whereas AT1 R167A mutation disrupted Ang II- and Hcy-induced β-arrestin 2 recruitment (Fig. 5f, g). Thus, Arg[167] is therefore a potential binding site for Hcy on the AT1 receptor and mediates both Ang II- and Hcy-induced AT1 receptor activation. To further distinguish the respective interaction of Arg[167] with Ang II and Hcy, AT1 R167N, and R167K mutations were applied in BRET assay. Consequently, R167K

mutation blocked both Ang II- and Hcy-induced β-arrestin 2 recruitment, whereas R167N mutation only abolished Hcy induced the β-arrestin 2 recruitment but not Ang II (Fig. 5f, g). These results indicated that the polar interaction of R167 with the Ang II is enough to support its downstream β-arrestin 2 recruitment, whereas both polar and basic properties of R167 is essential for its mediating Hcy-induced β-arrestin 2 signaling. These results are in consistent with our docking model that a salt

bridge formed between Hcy and Arg[167]. Together, the data shown here disclosed that the Arg[167] is the key residue responsible for both Ang II and Hcy interactions, but with distinct binding modes.

**S–S bonds do not mediate Hcy-induced AT1 receptor activation.** As Ang II and Hcy did not show typical competitive features

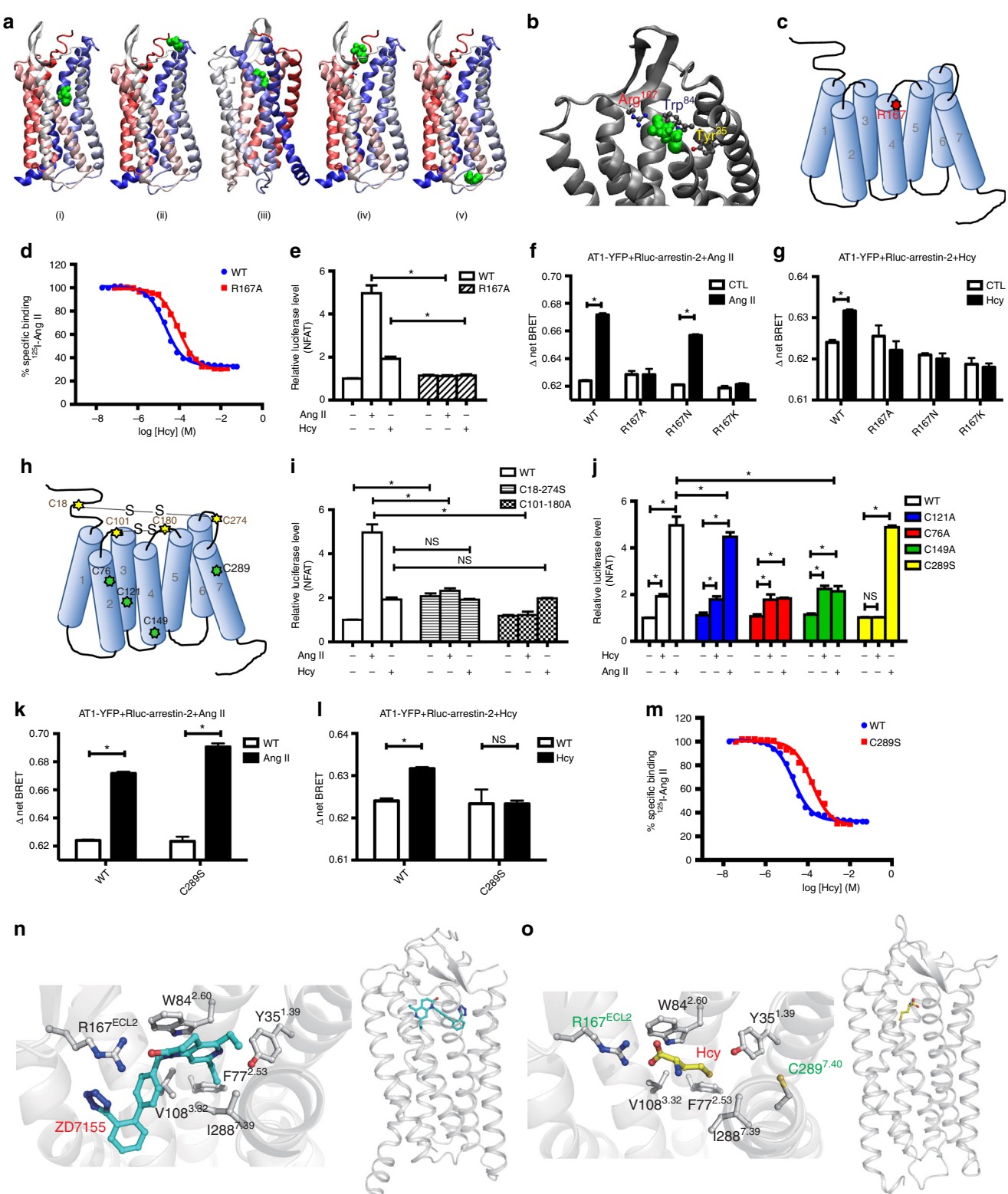

in radioligand binding assays, we wondered whether there are other binding sites on the AT1 receptor for Hcy. As Hcy has a very high propensity to form disulfide bonds (S–S bonds) with proteins[32], and the disulfide bonds of the AT1 receptor are reported to be regulated by thiol antioxidants[25], we speculated that Hcy might interact with the AT1 receptor disulfide bridges. The four cysteines in the extracellular loops (ECLs) of the AT1 receptor form two disulfide bridges (Cys[18] with Cys[274] and Cys[101] with Cys[180]), and these disulfide bridges have been reported to affect the binding affinity of Ang II and losartan, respectively[26,33]. Thus, the two cysteines forming one disulfide bond were mutated together, producing two mutants, each deficient in one disulfide bridge (Fig. 5h). The Cys[18] and Cys[274] mutation has been reported to induce constitutive activation of the AT1 receptor, while disruption of the Cys[101]–Cys[180] disulfide bridge also inhibits the Ang II binding affinity[33–35]. Of interest, although the activation by Ang II was blocked in both mutations, the effect of Hcy in inducing NFAT signaling was not affected (Fig. 5i). These data suggest that neither of the two disulfide bridges in the AT1 receptor mediated the Hcy-AT1 receptor binding.

**Cys[289] of AT1 mediates Hcy-induced AT1 receptor activation**. Hcy has a free sulfhydryl group and easily binds to other thiol-containing amino acids, such as the cysteines of fibronectin, and forms disulfide bonds[36]. Therefore, we further hypothesized that Hcy may bind to the cysteines of the AT1 receptor through the thiol groups. Accordingly, we mutated the cysteines that do not form disulfide bridges in the natural conformation of the AT1 receptor (i.e., Cys[121], Cys[76], Cys[149], and Cys[289]). As yet, there have been no reports regarding Cys[121] in AT1 receptor activation. The C121A mutation did not influence Ang II- or Hcy-induced NFAT activation (Fig. 5j). We did observe a marked decrease in Ang II-induced NFAT activation in C76A mutants, which is consistent with a previous report indicating that Cys[76] is one of the Ang II binding sites in the AT1 receptor[34] (Fig. 5j). However, Hcy-induced NFAT signaling was not affected in the C76A mutant. Cys[149] mutation has not yet been reported to regulate AT1 receptor activation. Of interest, the C149A mutant significantly reduced Ang II-induced NFAT activation but did not affect Hcy-induced NFAT activation (Fig. 5j). In contrast, the C289S mutation did not influence Ang II-induced AT1 activation, which is in accordance with a previous study[35], but completely blocked Hcy-induced NFAT activation (Fig. 5j) and β-arrestin 2 recruitment (Fig. 5k, l), indicating that Cys[289] specifically mediated the Hcy-induced AT1 receptor activation. To further confirm this observation, we conducted competitive radioligand

binding assays of the C289S-mutated AT1 receptor. The C289S mutation significantly inhibited Hcy-AT1 receptor binding (pK$_i$ of Hcy-WT AT1 vs. Hcy-C289S AT1: $4.678 \pm 0.024$ vs. $3.798 \pm 0.042$, mean $\pm$ SEM, Fig. 5m). The scientific literature indicates that mutation of Cys[289] does not influence [Sar[1], Ile[8]] Ang II-AT1 receptor binding[34]. Thus, Hcy and Ang II interact with the AT1 receptor via distinct binding sites, despite preserved overlaps. These results may explain the distinct conformations observed in the BRET assay after Ang II or Hcy administration (Fig. 4c). Of the sites studied above, Arg[167] and Cys[289] were predominantly responsible for Hcy-induced AT1 receptor activation.

We further performed a molecular dynamics simulation to better understand how Hcy activates the AT1 receptor via both the Arg[167] and Cys[289] sites. Molecular dynamics simulation revealed that Hcy was very close to both Arg[167] and Cys[289], as evidenced by the measured distances (Supplementary Fig. 15a, b). In cluster 1, Hcy stayed in the upper part of AT1 receptor, between TMD4 and TMD5 (Supplementary Fig. 16a). In cluster 2, Hcy was located to the top of AT1 receptor initially, and spontaneously moved to the middle-upper part of AT1 receptor (Supplementary Fig. 16b). In cluster 3, Hcy stayed at the initial binding site (Supplementary Fig. 16c). Hcy binds to a ligand binding pocket similar to the naphthyridin-2-one moiety of ZD7155 in the AT1 receptor/ZD7155 complex structure (Fig. 5n). The pocket was defined by the hydrophobic residues F77[2.53], V108[3.32], and I288[7.39] at the bottom and Y35[1.39] and W84[2.60] at the top. In our current model, which derived from the AT1 receptor crystal structure stabilized by an antagonist, R167[ECL2] forms a stable salt bridge with the carboxylic acid of Hcy whereas the distance of C289[7.40] and Hcy seems not to be close enough for a disulfide bond formation. Nevertheless, the position of Hcy would be dynamic in response to Hcy interaction, to shorten the distance between C289[7.40] and Hcy, which may subsequently form a disulfide bond (<6 Å; Fig. 5o). Moreover, in cellular assays, we observed different dynamics and efficacy of Ang II-induced PKC and ERK1/2 phosphorylation (Fig. 2a–e). Thus, a different but overlapping binding pocket, as well as different conformational changes of the intracellular domain of the AT1 receptor induced by Ang II and Hcy could account for their distinct properties in cellar signaling.

**Hcy and Ang II synergistically activate the AT1 receptor**. Because both Ang II and Hcy interdependently activate the AT1 receptor, we asked whether Hcy and Ang II synergistically activate the AT1 receptor. As shown by the Ca$^{2+}$ flux, Hcy (100 μM) markedly increased the maximum response and decreased the

**Fig. 5** The Arg[167] and Cys[289] residues are involved in Hcy-induced AT1 receptor activation. **a** Binding structures predicted in the global docking stage. **b** Binding mode of the AT1 receptor in cluster 3. Hcy is shown in green. Amino acid residues involved in ZD7155 binding are shown in gray. **c** Schematic diagram of the AT1 receptor Arg[167] mutation. **d** The specific binding of [[125]I]-Ang II and the WT AT1 receptor vs. the R167A-mutated AT1 receptor with different competing concentrations of Hcy. The data represent as mean $\pm$ SEM, $N = 6$. **e** Ang II (1 μM)- or Hcy (100 μM)-induced NFAT signaling in HEK293A cells transfected with the R167A-mutated AT1 receptor, detected by the dual-luciferase assay system (Promega). The data represent as mean $\pm$ SEM, $N = 6$, *$P < 0.05$, two-way ANOVA followed by the Bonferroni post hoc test. **f, g** The measurements of intermolecular Δnet BRET of AT1R and β-arrestin-2 in HEK293A cells (transfected separately with four AT1-YFP Arg[167] mutants and Rluc-β-arrestin-2) stimulated by Ang II (1 μM) or Hcy (100 μM). The data represent as mean $\pm$ SEM, $N = 6$, *$P < 0.05$, two-way ANOVA followed by the Bonferroni post hoc test. **h** Schematic diagram of the AT1 receptor disulfide bridge mutations and the Cys[121], Cys[76], Cys[147], Cys[289] mutations. **i, j** Ang II (1 μM)- or Hcy (100 μM)-induced NFAT signaling in HEK293A cells transfected separately with C18–274S, C101–180A (**i**), C76A, C121A, C149A, or C289S (**j**) mutated AT1 receptors, detected by the dual-luciferase assay system (Promega). The data represent as mean $\pm$ SEM, $N = 6$, *$P < 0.05$, two-way ANOVA followed by the Bonferroni post hoc test. **k, l** The measurements of intermolecular Δnet BRET of AT1R and β-arrestin-2 in HEK293A cells (transfected separately with two AT1-YFP Cys[289] mutants and Rluc-β-arrestin-2) stimulated by Ang II (1 μM) or Hcy (100 μM). The data represent as mean $\pm$ SEM, N = 6, *$P < 0.05$, two-way ANOVA followed by the Bonferroni post hoc test. **m** The specific binding of [[125]I]-Ang II and the WT AT1 receptor vs. the C289S-mutated AT1 receptor with different competing concentrations of Hcy. The data represent as mean $\pm$ SEM, $N = 6$. **n** Schematic diagrams of ZD7155-AT1 receptor interaction. **o** Schematic diagrams of Hcy-AT1 receptor interaction

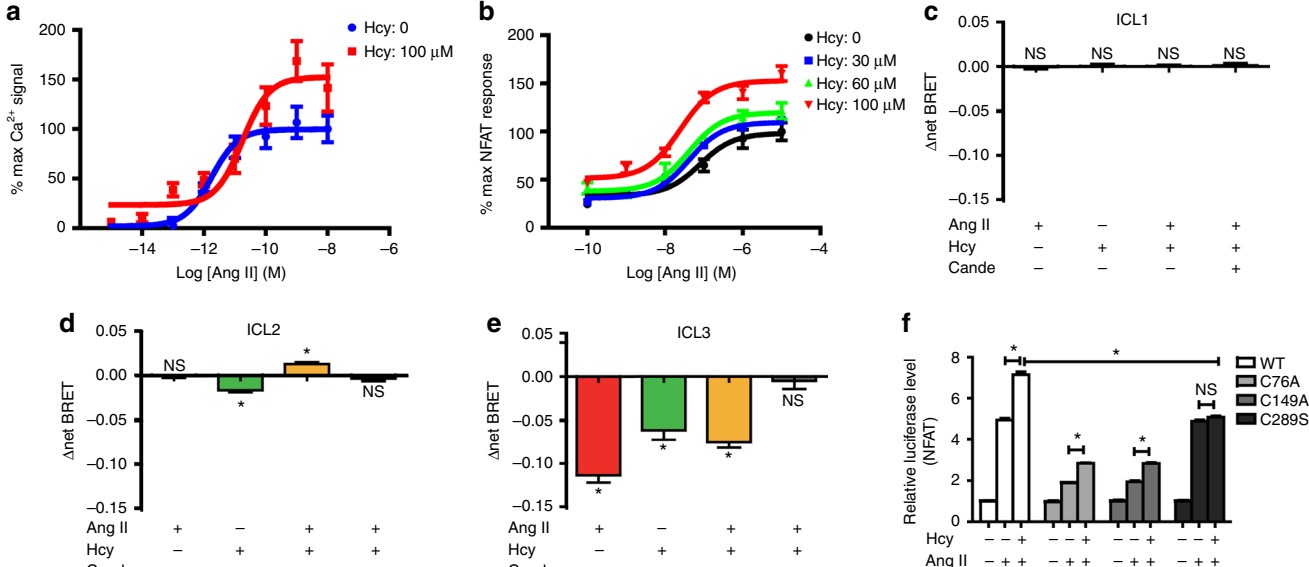

**Fig. 6** Hcy and Ang II synergistically activate the AT1 receptor. **a** Ang II-induced $Ca^{2+}$ signaling in the presence and absence of Hcy (100 μM), detected using Fluo 3-AM and quantified by the Leica confocal microscope system. The data represent as mean ± SEM, $N = 6$. **b** Ang II-induced NFAT signaling in the presence and absence of Hcy (30–100 μM), detected by the dual-luciferase assay system (Promega). The data represent as mean ± SEM, $N = 6$. **c**–**e** Intramolecular Δnet BRET in HEK293A cells (transfected with the three constructs ICL1–3) stimulated by Ang II (1 μM) plus Hcy (100 μM) with or without candesartan (0.1 μM) pretreatment. The data represent as mean ± SEM, $N = 6$, *$P < 0.05$, Student's $t$ test. **f** Ang II-induced NFAT activation with or without Hcy in HEK293A cells separately transfected with the C76A-, C149A-, or C289S-mutated AT1 receptor, detected by the dual-luciferase assay system (Promega). The data represent as mean ± SEM, $N = 6$, *$P < 0.05$, two-way ANOVA followed by the Bonferroni post hoc test

$EC_{50}$ of Ang II-induced AT1 receptor activation ($-Log[EC_{50}(M)]$) of Ang II vs. Hcy + Ang II: $11.8 \pm 0.18$ vs. $10.7 \pm 0.23$, $n = 3$ independent experiments; % maximum responses of Ang II vs. Hcy + Ang II: $99.7 \pm 5.03$ vs. $152 \pm 10.3$, Fig. 6a and Supplementary Table 7). Consistently, Hcy dose-dependently increased the maximum response of Ang II-activated NFAT signaling (Fig. 6b, Supplementary Table 8). We next utilized the intramolecular FlAsH BRET sensors of the AT1 receptor to examine the specific conformational changes induced by a synergistic effect of Hcy and Ang II. In contrast to the BRET sensors of ICL1 and ICL3, for which the synergistic application of Hcy and Ang II shows a similar Δnet BRET pattern compared to the application of either Hcy or Ang II individually, simultaneous administration of Hcy and Ang II caused a unique conformational change of ICL2 of the AT1 receptor, with an increased Δnet BRET (Fig. 6c, e). This result indicated that synergic application of Hcy and Ang II induced a unique conformation of the AT1 receptor, different from the conformation induced by either Ang II or Hcy alone. This effect was completely abolished by the AT1 receptor blocker candesartan. The observed conformational change specifically located in ICL2 of the AT1 receptor provided a primary structural insight into the allosteric regulatory role of Hcy in Ang II-induced AT1 receptor signaling.

Our cellular studies have indicated that $Arg^{167}$ is a shared binding site for both Hcy and Ang II, while $Cys^{289}$ is specific for the Hcy-AT1 interaction. We asked whether $Cys^{289}$ mediated the synergistic effect between Hcy and Ang II. Hcy (100 μM) significantly increased Ang II-induced NFAT activation in WT AT1 receptor-transfected groups. In contrast, the C289S mutation of AT1 completely abolished the Hcy-aggravated Ang II effect (Fig. 6f), while the C76A or C149A mutation did not, indicating that $Cys^{289}$ mediated the synergistic effect of Hcy and Ang II and that Hcy might allosterically interact with the AT1 receptor through $Cys^{289}$.

Conclusively, Hcy is an endogenous ligand and partial agonist of the AT1 receptor and directly activates the AT1 receptor via

the $Arg^{167}$ and $Cys^{289}$ sites. Hcy causes a distinct conformational change of the AT1 receptor compared to Ang II. Hcy and Ang II synergistically activate the AT1 receptor and aggravate aneurysmal vascular injury, which can be blocked by AT1a receptor deletion or ARB (Fig. 7).

## Discussion

In this study, we defined a novel mechanism by which HHcy exaggerates vascular injury in mice with AAA. Based on previous reports, the only endogenous ligand of the AT1 receptor is Ang II, although the AT1 receptor can be activated in the absence of Ang II by various stimulations, including mechanical stretch, AT1 receptor autoantibodies, and some artificial mutations of the AT1 receptor (constitutive activated mutations, CAM). In the present study, we have demonstrated that Hcy at concentrations indicative of mild to moderate HHcy directly binds to the AT1 receptor. Compared to Ang II-induced AT1 receptor activation, Hcy differs in various ways. First, the binding affinity of Hcy to the AT1 receptor is $10^{5}$ times lower than that of Ang II, whereas the plasma concentration of total Hcy is ~5 logs magnitude higher than Ang II. As only ~1% of total Hcy exists as free reduced Hcy[37], the other forms (protein-bound and oxidized) of Hcy may also take part in the interaction of Hcy and AT1 receptor as a partial agonist of AT1 receptor in vivo. Further investigations are required to clarify which forms of Hcy are active with respect to AT1 receptor interaction. Second, even at the saturated concentration, the Hcy-induced PKC and ERK phosphorylation was significantly lower than that observed at a saturated concentration of Ang II. The lowered efficacy of Hcy compared to Ang II toward the AT1 receptor suggested that it was a partial agonist. Third, Ang II and Hcy induced distinct conformational changes of the cytoplasmic domains of the AT1 receptor, particularly ICL2. This structural difference induced by Ang II and Hcy could explain the different observed patterns of AT1 receptor signaling stimulated by these ligands. Fourth, Hcy differs from Ang II with regard to

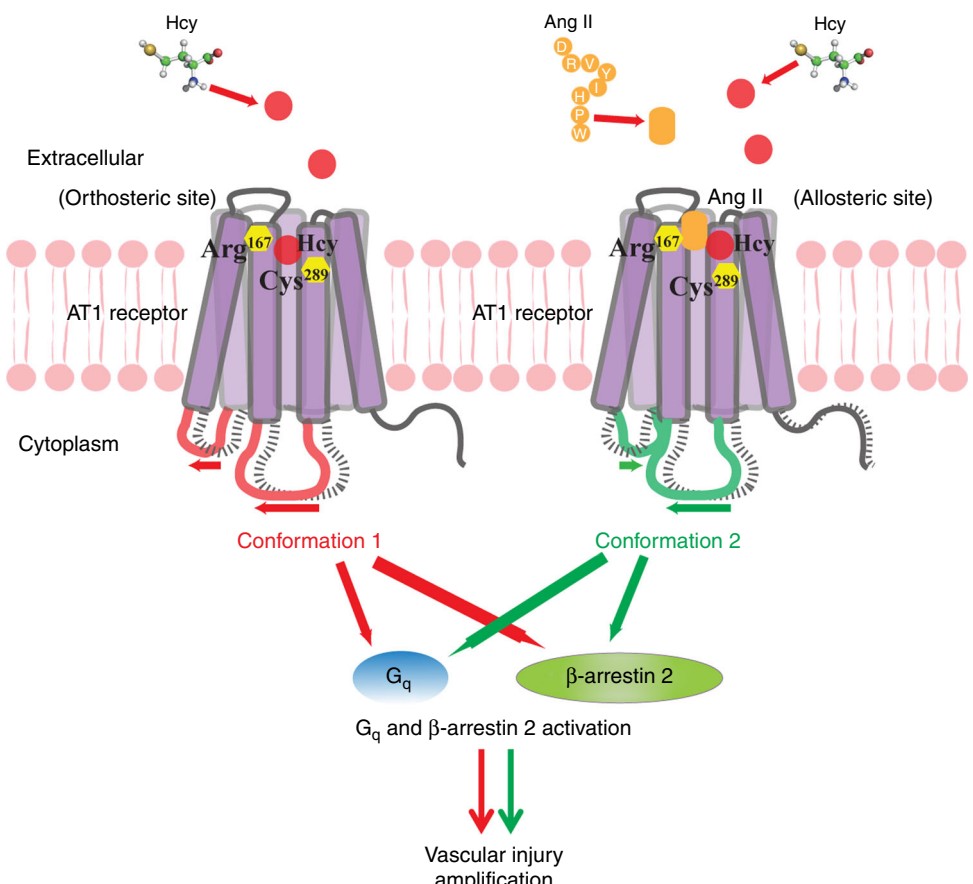

**Fig. 7** Schematic diagram of Hcy-induced AT1 receptor activation and the Hcy-Ang II synergistic effect on the AT1 receptor. Hcy is an endogenous ligand and partial agonist of the AT1 receptor and directly activates the AT1 receptor via the Arg[167] and Cys[289] sites. Hcy causes a distinct conformational change of the AT1 receptor compared to Ang II. Hcy and Ang II synergistically activate the AT1 receptor and aggravate vascular injury

binding sites on the AT1 receptor. A number of amino acids in the AT1 receptor, such as Asn[111], Arg[167], Lys[199], and Leu[222], are involved in the Ang II-AT1 receptor interaction[38]. In our current study, Arg[167] and Cys[289] were revealed as the two potential binding sites of Hcy on the AT1 receptor. Arg[167] overlapped with the Ang II binding sites, and this could potentially explain the partial inhibition observed in radiolabeled Ang II-AT1 binding assays. Moreover, the examination of the effects of a series of Arg[167] mutation on β-arrestin-2 recruitment induced by Ang II and Hcy, we demonstrated that the Arg[167] interact with Ang II and Hcy via distinct interaction modes. The other Hcy binding site, Cys[289], is unlikely to be a binding site of Ang II, due to the fact that a previous study indicated that [Sar[1], Ile[8]] Ang II bound to the WT and C289A mutant AT1 receptor with the same affinity and capacity[34]. It has also been reported that Cys[289] mediates stretch-induced AT1 receptor activation but not Ang II-induced AT1 receptor activation[35], which is consistent with our study. In summary, our study revealed Hcy as a novel endogenous ligand of the AT1 receptor, with distinct interaction modes compared to Ang II.

Another interesting finding is that Hcy may allosterically modulate the AT1 receptor and synergistically increase Ang II-induced vascular injury. Allosteric modulators interact with receptors via binding sites different from the orthosteric sites of its endogenous ligands. Therefore, the allosteric modulators can "fine-tune" the classical ligand-receptor responses and thus are potential pharmacological targets[39]. Agonistic autoantibodies of the AT1 receptor (predominantly found in preeclampsia or malignant hypertension patients) modulate the

AT1 receptor via ECL2, which is outside the orthosteric binding site of Ang II[11]. In the present study, Hcy binds to Cys[289] and causes a unique conformational change of ICL2 of the AT1 receptor, which is not involved in the binding site of Ang II, suggesting an allosteric modulation. Indeed, in the C289S mutation group, Hcy did not amplify Ang II-induced NFAT activation as it did in the WT group, indicating that Cys[289] is an allosteric binding site for Hcy and mediates the synergistic effect between Hcy and Ang II. Interestingly, a molecular dynamics study indicated that Hcy forms a salt bridge interaction with Arg[167], and the thiol group of Hcy exhibited the tendency to form a disulfide bridge with Cys[289], suggesting a dynamic interaction of Hcy with the AT1 receptor. Therefore, in theory, a small molecule targeting Cys[289] of the AT1 receptor could be designed as a new therapy to ameliorate Hcy-aggravated AAA.

To prevent or treat vascular injuries in patients with HHcy, including the *MTHFR* C677T population, the cystathionine-β-synthase (CBS)-deficient population, hypertensive patients with HHcy, and abdominal aortic aneurysm patients with HHcy, Hcy-induced AT1 receptor activation should be taken into consideration. Therefore, in addition to lowering total plasma Hcy, these patients may benefit from using AT1 receptor blockers rather than ACEIs, as our data showed that enalapril does not inhibit Hcy-induced AT1 receptor activation and HHcy-aggravated AAA formation, while telmisartan does. The particular activating pattern of Hcy and the clinical effects of ARBs in patients with HHcy remain to be elucidated.

## Methods

**Materials**. Angiotensin II (A9525), DL-homocysteine (H4628), enalapril (E6888), losartan (61188), telmisartan (T8949), rosiglitatone (R2408) and elastase (E1250) were purchased from Sigma-Aldrich (St. Louis, MO, USA). Candesartan was purchased from Takeda Pharmaceutical Co. Ltd. (Osaka, Japan). Antibodies against phosphorylated and total extracellular signal-regulated kinase (p-ERK1/2, t-ERK1/2) and phosphorylated and total protein kinase C (p-PKC, t-PKC) were purchased from Cell Signaling Technology (Boston, MA, USA). The interleukin-6 (IL-6) and monocyte chemoattractant protein 1 (MCP-1) ELISA kits were from BioLegend (San Diego, CA, USA). Fluo-3-AM molecular probes (F14218) were purchased from Molecular Probes (Eugene, OR, USA). The Dual-Luciferase Reporter Assay System (E1910) was from Promega (Madison, WI, USA). Iodine[125] (I100813L) was from PerkinElmer (Boston, MA, USA).

**Animal treatments**. All animal studies and experimental procedures were approved by the Institutional Animal Care and Use Committee (IACUC) of Peking University Health Science Center (LA2015–142), an ethical organization authorized by Beijing Municipal Science and Technology Commission. C57BL/6J mice (male, 8 weeks) were purchased from Vital River (Beijing, China). $AT1a^{-/-}$ mice (male, 8 weeks) were obtained from The Jackson Laboratory (US). The same genotypic mice were randomly divided into groups of different treatments using the online randomization (http://www.graphpad.com/quickcalcs/index.cfm). The sample size in animal study was estimated using Akaike information criterion.

As male mice are more susceptible to AAA than female mice[40,41], male mice were applied for AAA induction. Male C57BL/6J mice at 8 weeks of age were fed a normal mouse chow diet supplemented with or without 1.8 g/L DL-homocysteine daily-prepared in drinking water for 4 weeks[4]. Elastase-induced AAA was performed in mice accordingly[42]. The mice were anaesthetized, and the infrarenal aortas were exposed, isolated and wrapped with sterile cotton, followed by addition and soak of 20 µl elastase (68.68 U/ml) for 40 min. Then, elastase-soaked cotton were removed, and 0.9% NaCl was used to perfuse the abdominal cavities before suturing. Aortas were collected 14 days after surgery. CaPO4-induced AAA was performed in mice as previously described[43]. In brief, mice were anaesthetized, and the infrarenal aortas were exposed, isolated and wrapped with gauze presoaked in 0.5 M CaCl2 for 10 min. Then, the CaCl2-soaked gauze was removed, and the aortas were wrapped with gauze presoaked in PBS for another 5 min to form CaPO4. The abdominal cavities were washed with 0.9% NaCl before suturing. The aortas were collected 14 days after surgery. For the telmisartan group, the mice were supplemented with 10 mg/kg per day telmisartan in drinking water for 4 weeks as previously described[44]. For the enalapril group, the mice were supplemented with estimated 30 mg/kg per day enalapril in drinking water (0.15 g/L) for 4 weeks from 8 weeks of age[45]. The dose was chosen to achieve a profound effect on the inhibition of angiotensin I-elevated blood pressure in mice, as previously described[46]. In the above two models, the systolic blood pressure was monitored by tail-cuff plethysmography before and after the AAA induction.

**Tail-cuff measurement of systolic blood pressure**. A non-invasive computerized tail-cuff system (BP-98A, Softron Biotechnology) was used to measure the systolic blood pressure (SBP) of mice[47,48]. The equipment was kept clean free from foreign scent and blood odor. The investigator was blinded with respect to the experimental groups to perform the measurements, while mice were tested in a randomized fashion. The mice underwent 7 consecutive days of training sessions from 1 to 5 PM each day to become accustomed to the tail-cuff procedure. Sessions of recorded measurements were then made by a single investigator at the same time on 3 consecutive days. Five measurements were daily performed on each mouse, so that the average of total of 15 measurements was represented as the SBP of each mouse.

**Analysis of plasma lipid levels**. Plasma triglyceride and total cholesterol levels were assayed with kits from Zhong Sheng Bio-technology (Beijing).

**Analysis of aneurysmal morphology in mice**. AAA mice were killed, the whole aortas including thoracic and abdominal segments were isolated, and perivascular connective tissues were removed. Dissected aortas were fixed on a black wax surface in a dissecting pan for photography with a dividing ruler, and the diameters of the aortas were measured using Image-Pro Plus 6.0 with the ruler. For Verhoeff–Von Gieson staining and Gomori staining, infrarenal aortas were perfusion-fixed with 4% paraformaldehyde. Serial frozen sections (7 µm thick, ~1 mm apart) were generated from the infrarenal aorta of each mouse. Sixteen serial sections for each mouse were analyzed by Verhoeff–Von Gieson staining or Gomori staining for elastin assessment. The elastin degradation was graded on a scale of 1–4, where 1—<25% degradation, 2—25 to 50% degradation, 3—50 to 75% degradation, and 4—>75% degradation[49]. The data were evaluated blindly by two independent investigators and presented as the mean of 16 serial sections for each mouse.

**Gelatin zymography**. Abdominal aortic rings treated with Hcy (100 µM) in vitro were incubated in culture medium for 20 h. The conditioned medium was electrophoresed on 10% SDS-polyacrylamide gels containing 1.0 mg/ml gelatin (Sigma-Aldrich). The gels were washed twice in 2.5% Triton X-100, incubated for 48 h (37 °C) in zymography buffer (50 mM Tris-HCl (pH 7.5), 150 mM NaCl, 10 mM CaCl2, and 0.05% sodium azide) and stained with Coomassie brilliant blue.

**Measurements of Ang II concentration and renin activity**. Ang II concentrations in mouse plasma and culture medium were measured using an [125I]-Ang II radioimmunoassay kit (Bühlmann Laboratories, Basel, Switzerland)[50]. For plasma detection, dimercaprol (1.6 mM) and 8-quinolinol hemisulfate (3.4 mM) were added to the plasma immediately after blood withdrawal to avoid Ang I–Ang II transition. The samples, calibrators, and controls were first preincubated for 16 h with an anti-Ang II antibody. [125I]-Ang II was then added and competes with Ang II present in samples, calibrators and controls for the same antibody binding sites in a second 6 h incubation step. After this incubation, a solid-phase second antibody was added to the mixture. The antibody-bound fraction was precipitated and counted in a gamma counter. The radioactive values in calibrators were regressed with respective Ang II concentrations into the standard curve. Based on it, the concentrations of Ang II were calculated in samples.

For evaluation of the plasma renin activity, the blood sampling was completed within 20 s to minimize stress-induced renin secretion. The plasma was separated by centrifugation at 3000×g for 5 min at 4 °C. The plasma renin activity was determined using a modification of the technique as previously described[51]. Angiotensin I was generated in vitro during a 1-h incubation at pH 6.5 and 37 °C; the incubation mixture contained 50 µl of test plasma, 10 µl of 2 M maleic acid buffer (pH adjusted to 6.5 with NH4OH), 1 µl of British anti-lewisite (BAL) (1.7 g/100 ml), and 1 µl of 8-hydroxyquinoline (6.6 g/100 ml). The 50 µl of aliquot plasma without incubation was applied as basal control. Angiotensin I was then measured with radioimmunoassay kits (NEN-DuPont, Boston, MA, USA) in incubated samples and basal control. The generated Angiotensin I in test samples subtracted with basal control was represented as plasma renin activity.

**Analysis of Hcy concentration**. The concentrations of total Hcy, including free thiol and disulfide form of Hcy, in mice plasma were measured using enzymatic cycling method through HITACHI LAbOSPECT biochemical analyzer in Peking University People's Hospital[52]. Following addition of reaction reagents, disulfide form of Hcy was firstly reduced by trichloroethyl phosphate into free thiol Hcy. The total free thiol Hcy then reacted with SAM to form SAH. SAH hydrolyzed by SAHase produced adenosine. Adenosine was further hydrolyzed into NH4+ and hypoxanthine. Adenosine-derived NH4+ finally reacted with NADH. The reacted NADH measured by the biochemical analyzer was proportional to Hcy amount, and the concentrations of plasma Hcy were calculated based on the amount of reacted NADH. Free thiol Hcy in mice plasma was measured through gas chromatography-mass spectrometer.

**Cell culture**. The human embryonic kidney cell line HEK293A (R70507) were purchased from Thermo Fisher Scientific Inc (Waltham, MA, USA). The African green monkey kidney fibroblast cell line COS7 (ATCC CRL-1651) were purchased from the American type culture collection (Manassas, VA, USA) and cultured in DMEM (high glucose) supplemented with 10% fetal bovine serum at 37 °C in a humidified atmosphere containing 5% CO2. The cells were confirmed of no mycoplasma contamination before applied.

**Quantitative real-time PCR**. Real-time PCR was performed on an Mx3000P qPCR system (Agilent, Santa Clara, CA, USA) with TransStart Top Green qPCR SuperMix (AQ131–04, TRANSGEN, Beijing, China). All samples were normalized to β-actin. The primer sequences for real-time PCR are shown in Supplementary Table 9.

**Western blot analysis**. The cell lysates were resolved by 10% SDS-PAGE for western blot analysis. The blots were subsequently incubated with primary antibodies (p-PKC, 1:1000, Cell Signaling Technology, 9371 s; t-PKC, 1:1000, Abcam, ab23511; p-ERK1/2, 1:1000, Cell Signaling Technology, 9101s; t-ERK1/2, 1:1000, Cell Signaling Technology, 9102s) and IRDye-conjugated secondary antibodies (1:10,000, Rockland Inc. Gilbertsville, PA, USA). The immunofluorescent signals were detected using Odyssey infrared imaging (LI-COR Biosciences, Lincoln, NB, USA). For the detection of phosphorylated proteins, phosphatase inhibitor cocktail tablets (Roche, Berlin, Germany) were added to the lysis buffer. The uncropped scans of all western blots were showed as Supplementary Fig. 17.

**Radioactive ligand binding assay**. The cell membranes were isolated from HEK293A cells (transfected with the human AT1 receptor or not transfected) or mice abdominal aorta using a lysis buffer containing 150 mM NaCl, 0.1% Triton X-100, 50 mM Tris-HCl, and 1 mM EDTA and supplemented with a protease inhibitor cocktail containing aprotinin, leupeptin, and pepstatin (pH 7.4). The cell lysate was homogenized with a 27-G needle and centrifuged at 12,000×g at 4 °C for 10 min. The supernatant was collected and further centrifuged for 60 min, and the precipitated membrane extracts were resuspended in a buffer containing 150 mM NaCl, 10 mM MgCl2, 0.5% SDS, 1% Triton X-100, 50 mM Tris-Cl, and 1 mM EDTA and supplemented with the protease inhibitor cocktail (pH 7.4).

For the binding assay, the membrane extracts were incubated at 4 °C with increasing concentrations of [$^{125}$I]-Ang II (0–100 nM) for 24 h. To remove the unbound [$^{125}$I]-Ang II, the mixture was incubated with 25% polyethylene glycol (PEG)–4000 for 30 min at room temperature and centrifuged at 3500 r.p.m. for 15 min. The membrane extracts were transferred to polystyrene tubes, and radioactivity was measured with a gamma counter (r911, Industrial General Corporation of University of Science and Technology of China). Non-specific binding (parallel tubes with non-transfected HEK293A cell membrane) was subtracted from the data. The binding data were normalized to protein amounts. The dissociation constant $K_D$ value and the receptor density $B_{max}$ value were determined.

**Measurement of the intracellular free calcium concentration (Ca$^{2+}$).** The intracellular Ca$^{2+}$ concentration was measured using Fluo-3-AM (Molecular Probes). HEK293A cells seeded in a confocal dish were washed three times with PBS and incubated with 10 μM Fluo-3-AM for 30 min at 37 °C in 5% CO$_2$. After proper rinsing, Ca$^{2+}$ fluorescence (488 nm) was recorded by confocal microscopy. Images were captured every 10 s after Ang II or Hcy stimulation. The images were quantified by a Leica Microsystems laser scanning confocal microscope (LAS AF-TCS SP5).

**NFAT dual-luciferase reporter assay.** The NFAT (Nuclear Factor of Activated T cells) Dual-Luciferase Reporter Assay downstream of the AT1 receptor was performed as previously described[53]. In brief, HEK293A cells were transfected using jetPEI DNA Transfection Reagent (Polyplus) in 35 mm dishes with 0.8 μg of the AT1 (human)-mCherry plasmid, 0.8 μg of the pGL3-NFAT luciferase plasmid or pGL3-Basic luciferase plasmid, and 0.08 μg of the pRL-TK plasmid[53]. The cells were cultured for 40 h after transfection, followed by the addition of Ang II or Hcy. The cells were further cultured for 6–8 h and collected using passive lysis buffer. NFAT activity was quantified by dual-luciferase reporter gene assays (Promega) following the instructions of the manufacturer.

**Molecular docking and molecular dynamics.** To identify the possible binding sites for Hcy on the AT1 receptor, we conducted molecular docking experiments as previously described[54]. The crystal structure of the human AT1 receptor (PDB code: 4YAY) was used[30]. The AutoDock 4.2 package was used in this work.

A molecular dynamics (MD) simulation was performed to investigate the interaction of Hcy with the AT1 receptor using the NAMD2 package[55]. The protein was embedded in the phospholipid bilayer. In addition, the system was first equilibrated for 10 ns, followed by a 100 ns production run. The MD simulation was carried out in the NPT ensemble with 1 bar pressure and a temperature of 300 K. The temperature and pressure of the system were maintained using the Nosé–Hoover thermostat and Langevin piston barostat method, respectively[56].

**Intramolecular FlAsH BRET.** HEK293 cells were transfected using PEI DNA Transfection Reagent in 6-well plates with 2 μg of pcDNA3-AT1 receptor-ICL-FlAsH-Rluc constructs. Twenty-four hours later, the pcDNA3-AT1 receptor-ICL-FlAsH-Rluc-transfected cells were resuspended and seeded into 96-well plates. After another 24 h, the cells were washed with Opti-MEM and incubated with 2.5 μM FlAsH-EDT$_2$ using a TC-FlAsH II In-Cell Tetracysteine Tag Detection Kit (Thermo Scientific, T34561) at room temperature for 30 min. The FlAsH-labeled cells were then washed twice with 1X BAL buffer from the TC-FlAsH kit. Ang II (1 μM) and Hcy (100 μM) with coelenterazine-h (5 μM) were added, and the BRET (Bioluminescence Resonance Energy Transfer) signals were determined on a Mithras LB 940 Multimode Microplate Reader with 492 nm excitation and 530 nm emission filters. The Δnet change in intramolecular FlAsH BRET ratio for each AT1 receptor-ICL-FlAsH construct was calculated by subtracting the BRET ratio measured after stimulation with vehicle in the same experiment.

**BRET measurement.** The AT1a-YFP and β-arrestin-2-RLuc plasmids were transiently transfected into HEK293T cells. Forty-eight hours after transfection, the cells were washed with PBS, detached with PBS and 5 mM EDTA, and resuspended in PBS with 0.1% (w/v) glucose at room temperature. The cells were then distributed (80 μg of protein per well) in a 96-well microplate (Corning Inc., Corning, NY, USA) and incubated in the presence or absence of Ang II (1 μM) or Hcy (100 μM) for 1 min. BRET between RLuc and YFP was measured after the addition of the RLuc substrate coelenterazine 400a (5 μM, Interchim) under a Thermo plate reader. The BRET signal was calculated as the ratio of emission of YFP (527 nm) to RLuc (370–480 nm).

**Statistical analysis.** All data are presented as the mean ± standard error of mean (SEM). GraphPad Prism 6.0 (GraphPad Software, San Diego, California, USA) was used in statistical analyses. For the statistical comparison, whether data are normally distributed was first evaluated. Then Brown–Forsythe test was made for checking similar variances among normally distributed data followed by Student's $t$ test for two-group comparisons and ANOVA for more than two-group comparisons if evaluation of similar variances was passed. Non-parametric tests were used where data were not normally distributed. In all cases, statistical significance

was concluded where the two-tailed probability was <0.05. Comparison of time-course MCP-1 and IL-6 secretion, AT1a, AGT, and ACE mRNA expression and Ang II release were analyzed using a two-way ANOVA followed by Bonferroni's multiple comparison tests. Comparisons of the $K_D$ and $B_{max}$ values, Ca$^{2+}$ signaling and NFAT luciferase assays were analyzed using an unpaired two-tailed Student's $t$ test. The data from the abdominal aortic aneurysm diameter showed a non-normalized distribution. Comparisons of the abdominal aortic aneurysm diameter were analyzed by a non-parametric Kruskal–Wallis test followed by Dunn's test. In all cases, statistical significance was concluded where the two-tailed probability was <0.05.

**Data availability.** All of the data supporting the findings in this study are available from the corresponding authors upon request.

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

## Acknowledgements

This research was supported by funding from the National Natural Science Foundation of the P. R. China (NSFC, 91539203, 81730010); National Key R&D Program of China (2016YFC0903000); International Cooperation and Exchanges NSFC (81220108004); the 111 Project of Chinese Ministry of Education (No. B07001); the National Program on Key Basic Research Projects (973 Program) (2012CB518002); and the National Science Fund for distinguished Young Scholars (81225002).

## Author contributions

T.L., B.Y., and Z.L. contributed equally to performing most experiments and writing manuscripts. J.L. and X.W. performed molecular docking experiments. M.Z. and H.Y. performed the measurement of homocysteine concentration. Y.W., M.M., and F.Y. performed the partial experiments. Y.F. performed partial experiments, wrote and revised manuscript and provides some scientific suggestion. X.W. and X.F. provided some scientific suggestion. J.S and W.K. hold the funding, designed the experiments and wrote and revised manuscript.

## Additional information

**Competing interests:** The authors declare no competing financial interests.

