## [Peer Review File · Nature Communications]

Reviewers' comments:

Reviewer #1 (Remarks to the Author):

Using mouse aortic aneurysm models, the authors demonstrate that hyperhomocysteinemia (induced by supplementation of drinking water with homocysteine) aggravates AAA formation through a mechanism that is dependent on AT1. The in vivo parts of the manuscript are convincing and mechanistic, although somewhat incremental.

For the radioactive ligand binding studies, were non-transfected HEK293 cells (e.g. cells that do not express AT1) used to control for non-specific binding? This might be particularly important to confirm that homocysteine actually has both orthosteric and allosteric effects on Ang II binding. There are discrepancies between the Methods (page 7), the Results (page 14), and Online Figure 1 with regard to the age of the mice that were given homocysteine-water (8 weeks or 12 weeks?).

How was plasma homocysteine measured, and does the assay distinguish between free thiol and disulfide forms of homocysteine?

Why were only male mice studied?

A major limitation of the cell culture experiments is the use of a relatively high concentration (100 μ M) of the free thiol form of D, L-homocysteine. This is about 100-fold higher than the plasma concentration of the free thiol form of homocysteine, even in patients with moderate hyperhomocysteinemia. Another limitation is that the K_i for Hcy binding is 5 logs higher than the K_i for Ang II binding. Together, these limitations raise the possibility that the in vivo effect of hyperhomocysteinemia may be unrelated to the partial agonist activity of Hcy observed in vitro.

The conclusion that Arg167 is a homocysteine binding site is not adequately supported by the data. The R167A mutation had a minimal effect on Hcy binding (Figure 5D). Furthermore, since the mutation caused complete loss of AT1 signaling to both Hcy and Ang II, it is not correct to conclude that it is a specific Hcy binding site.

The results with the C289S mutation are more convincing. Do the authors have any data to suggest that homocysteine forms a disulfide with Cys289?

Reviewer #2 (Remarks to the Author):

Hhcy has been previously shown to augment AAAs. The major conclusion of this manuscript is that Hhcy promotes AAA via direct stimulation of AT1a receptors. The reviewer acknowledges the structural studies demonstrate an ability of Hhcy to directly interact and stimulate this receptor. Also there is data to demonstrate that the augmented AAAs in Hhcy mice is inhibited by the absence of AT1a receptors. However, there is no direct evidence presented to demonstrate in vivo that Hhcy promotes AAA through direct AT1aR stimulation. It is surprising that this data has not been included.

Comments

1. The major conclusion of this manuscript is that AT1a receptors are stimulated independent of angiotensin II to promote AAA. For this statement to be included, there must be some study in which AAA are formed in mice that have had a manipulation to reduce the production of angiotensin II. For example, administration with an ACE inhibitor.

2. In figure 1, data represented in figure 1B is that maximal aortic diameters were approximately 1 mm. Most measurements in this region for normal aorta would be in the 0.4 to 0.5 mm range.

would be helpful to provide the starting size of each of these aortas. Based on these measurements, the infra renal aorta of all 4 groups is greatly expanded, with augmentation in the Hhcy group that are wild type for At1a receptors. However, Figure 1C and D seems to indicate there are minimal elastin breaks in 3 groups. How can this be reconciled?

3. The authors cited a reference (25) that suggests potential effects of AT1aR in the aortic wall contributes to AAA to support their hypothesis. However, two groups have demonstrated that deletion of AT1a receptors in smooth muscle cells has minimal effects on aortic pathology in angiotensin II infused mice. Therefore, there should be additional information demonstrating that this receptor type in smooth muscle cells is relevant to the disease process. In the studies represented in online figure 4, aortic rings were incubated with Hhcy and effects were determined. For this to be meaningful, there has to be additional data demonstrating similar changes during incubation with angiotensin II.

4 Telmisartan was used in most experiments to block AT1R, with a single exception that candesartan was used without providing rationale why either was used. More importantly, it is well-known that telmisartan has strong PPARgamma activation effect. Therefore, data interpretation should at least discuss the potential off-target effects.

5. Online Figure 5: The authors performed ex vivo experiment and stated that AngII secretion from aortic ring explants was not altered by homocysteine within 12 hours, which does not rule out increase of AngII after 12 hours. Additionally, the authors measured mRNA of AGT, ACE, and AT1aR, but not renin, in the aortic explant. If renin is not present in the aortic ring, how was AngII produced?

6. No immunofluorescent staining method is described. The specificity of all antibodies needs to be validated.

7. There is considerable skepticism about the accuracy of measuring angiotensin II. The authors have used this using a kit in which the reviewer can find no information. It is important to have a full description of this assay. For these measurements to be generally accepted, it would be helpful to provide validation studies.

8. It should be stated where the aortic rings are derived from. If these rings are derived from regions that are aneurysm-resistant, the relevance of findings in these tissues must be discussed.

9. On page 30 the last paragraph, the authors state that "Therefore, in addition to lowering total plasma Hcy, these patients may benefit from using AT1 receptor blockers rather than ACEIs, as our data showed that enalapril does not inhibit Hcy-induced AT1 receptor activation, while telmisartan does." There has no data to support this conclusion.

Reviewer #3 (Remarks to the Author):

This is a review of the computational portions of the paper. The authors used small molecule docking with AutoDock 4.2 to identify potential homocysteine binding sites on AT1 using both global docking and local refinement. AT1 is a membrane-embedded GPCR, so docking to lipid-facing regions is unusual, and any predicted conformations in the lipid bilayer region would be suspect. Fortunately, only cluster 1 looks to suffer from this problem. The most interesting cluster (cluster 3) was in the solvent-facing surface of AT1, so this is a reasonable prediction. The follow-up experiments support the binding location and the identification of Arg167 as a key binding residue. Additionally, MD simulations of the HCY at cluster 3 were stable.

The authors also suggest that HCY may make a disulfide bond with C289. In Fig 5J, these atoms are not close enough to bond, but it is conceivable that the HCY position would be dynamic enough

to allow bonding.

Overall, this paper uses fairly standard computational methods in reasonable manners to support their stated conclusions.

One minor edit on p. 23 line 22: "stimulation" should be "simulation".

Reviewers' comments:

Reviewer #1 (Remarks to the Author):

Using mouse aortic aneurysm models, the authors demonstrate that hyperhomocysteinemia (induced by supplementation of drinking water with homocysteine) aggravates AAA formation through a mechanism that is dependent on AT1. The in vivo parts of the manuscript are convincing and mechanistic, although somewhat incremental.

Response: *Many thanks for the reviewer's positive comments.*

For the radioactive ligand binding studies, were non-transfected HEK293 cells (e.g. cells that do not express AT1) used to control for non-specific binding? This might be particularly important to confirm that homocysteine actually has both orthosteric and allosteric effects on Ang II binding.

Response: *Following reviewer's suggestion, we performed the radioactive ligand binding assay using non-transfected HEK293 cells as non-specific binding controls. We subtracted the non-specific bindings from the corresponding binding of AT1-overexpressing HEK293 cells and recalculated K_i and K_D respectively. The final results were shown in Figure 3A-B, 5D and 5I in the revised manuscript. Our new data reinforced the notion that homocysteine has both orthosteric and allosteric effects on Ang II binding.*

There are discrepancies between the Methods (page 7), the Results (page 14), and Online Figure 1 with regard to the age of the mice that were given homocysteine-water (8 weeks or 12 weeks?).

Response: *We are sorry about the editing error in the previous version of our manuscript. The mice were given homocysteine-water at 8 weeks of age. We modified the age of mice in the part of "Animal Treatments" on page 26 of the Methods section in the revised manuscript.*

How was plasma homocysteine measured, and does the assay distinguish between free thiol and disulfide forms of homocysteine?

Response: *In our revised manuscript, we measured the total Hcy (including both free thiol and disulfide forms of Hcy) and free thiol Hcy in mice plasma using two different assays, which are Enzymatic Cycling assay and Gas Chromatograph-Mass Spectrometry respectively. (1) The concentrations of Hcy (including both free thiol and disulfide forms of Hcy) in mice plasma were measured using Enzymatic Cycling method through HITACHI LABOSPECT biochemical analyzer in Peking University People's Hospital (Cardiovasc Res. 2013;97(2):349-59). Following adding reaction reagents, disulfide form of Hcy was firstly reduced by trichloroethyl phosphate, which produce free thiol Hcy. The total free thiol Hcy then reacted with SAM to form SAH. Then SAH was hydrolyzed by SAHase to produce adenosine. Adenosine was further hydrolyzed into NH_4^+ and hypoxanthine. Adenosine-derived NH_4^+ finally reacted with NADH. The reacted NADH measured by the biochemical analyzer was proportional to Hcy amount, and the concentrations of plasma Hcy were calculated based on the amount of reacted NADH. Although multiple steps were involved in the enzymatic cycling reaction, the reaction buffers were added together, which enable the sequential proceeding of the reduction of disulfide form Hcy and subsequent reactions. Thus, this method was not sufficient to distinguish the free thiol and disulfide forms of Hcy, but only determinate the total Hcy. (2) In a separate experiment, we measured free thiol Hcy in mice plasma via Gas Chromatograph-Mass Spectrometry. Please refer to revised Online Table 2. Our results are consistently with what previously reported (Compr Physiol. 2015;6(1):471-505.), only approximate 1% of total Hcy exists as free reduced Hcy. We also modified the description of "Analysis of Hcy Concentration" part on page 29 of revised manuscript.*

Why were only male mice studied?

Response: *Epidemiological studies have demonstrated that men are more likely to suffer from AAA than women (Thorac Cardiovasc Surg 2013;61:15–21; J Vasc Surg 1997;25(3):561–568). Similar to human, male mice are more susceptible to AAA than female mice, as evidenced by that female mice exhibit far lower incidences of angiotensin II- or elastase-induced AAAs than males (Arterioscler Thromb Vasc Biol.*

2004;24:2116-2122; *Circulation*. 2017;135(4):379-391). Thus, we preferred male mice in induction of AAA *in vivo*. We added this rationale in “Animal Treatments” part of revised Method on Page 26 as well.

A major limitation of the cell culture experiments is the use of a relatively high concentration (100 μM) of the free thiol form of D, L-homocysteine. This is about 100-fold higher than the plasma concentration of the free thiol form of homocysteine, even in patients with moderate hyperhomocysteinemia.

Response: *In our experiments, the actual free thiol form of Hcy is approximately 1 μM in spite we administrate 100 μM Hcy in the cell culture medium, suggesting that Hcy concentration we applied in this study is within the pathological range. **Following is our detailed explanation:** Under normal conditions, the concentration of Hcy ranges from 5-10 μM in plasma, not exceeding 15 μM . An elevation of plasma Hcy (>15 μM) is manifested as hyperhomocysteinemia (HHcy). These concentrations mentioned above indicate the total Hcy including free thiol and disulfide forms in plasma. Moreover, only approximate 1% of total Hcy exists as free reduced Hcy. Thus, for moderate HHcy (30-100 μM), the concentration of free thiol Hcy should be around 0.3-1 μM . To investigate whether the Hcy concentration used in our cellular experiments are reasonable, we measured the concentration of free thiol Hcy in conditional medium following 100 μM Hcy treatment in various time points (1 minutes, 15 minutes and 8 hours) through GC-MS. Interestingly, the concentration of free thiol Hcy was about 1 μM within 1 minutes after adding 100 μM Hcy. This significant drop of free thiol Hcy in conditional medium may be due to oxidation of Hcy into disulfide form, uptake by cells or fast degradation of Hcy. So the actual concentration of free thiol Hcy in cell culture medium (~1 μM) is close to the pathologic concentration. Please refer to the responding Figure below.*

Another limitation is that the K_i for Hcy binding is 5 logs higher than the K_i for Ang II binding. Together, these limitations raise the possibility that the in vivo effect of hyperhomocysteinemia may be unrelated to the partial agonist activity of Hcy observed in vitro.

Response: *In vivo and within the pathological context, the plasma concentration of Hcy is 5 logs magnitude higher than AngII. Following is the detail of our explanation:* The concentrations of Ang II in human plasma varied 5-20 pM under physiological condition, and significantly elevated up to 300 pM in some pathological conditions, such as ketoacidosis, hypertension (European Journal of Clinical Investigation 1971;2(1):32-38; J Hum Hypertens. 2017;31(7):457-461). As we know that HHcy is defined as plasma Hcy >15 µM, which is more than 10^5 times as Ang II concentration even under pathological condition. In our current study, the HHcy mice plasma Hcy and Ang II concentrations are 20-30 µM and 300-500 pM respectively (Online Table 2-4). Thus, the marked distinct of concentrations between Hcy and Ang II assures the possibility of Hcy working as a partial agonist of AT1 receptor in vivo, although the K_i for Hcy binding is 5 logs higher than the K_i for Ang II binding. We also added this rationale in last two lines on page 21 of the revised manuscript.

Moreover, we performed new saturation binding assays with aortic membrane fraction of WT and AT1a^{-/-} mice with or without HHcy to confirm the direct interaction of Hcy and AT1a receptor in vivo. As a consequence, HHcy attenuated the B_{max} value but increased the K_D value of [¹²⁵I]-Ang II in interaction with WT aortic membrane fraction (B_{max} of control vs. Hcy: 26268 ± 697 vs. 22315 ± 618 fmol·mg⁻¹ protein

**P<0.05, non-specific binding was subtracted; KD of control vs. Hcy: 36.4 ± 4.6 vs. 43.2 ± 3.9 nM, Figure 3C), whereas AT1a knock out decreased both B_{max} and K_D values as well as disrupted the effect of HHcy on B_{max} and K_D values of [¹²⁵I]-Ang II binding with aortic membrane fractions. The HHcy-induced B_{max} attenuation at the saturated concentration of Ang II suggested that Hcy directly interact with the aortic AT1a receptor in vivo and in physiological receptor concentrations. Please refer to revised Figure 3C and paragraph 2 on page 12 of revise manuscript.*

The conclusion that Arg167 is a homocysteine binding site is not adequately supported by the data. The R167A mutation had a minimal effect on Hcy binding (Figure 5D). Furthermore, since the mutation caused complete loss of AT1 signaling to both Hcy and Ang II, it is not correct to conclude that it is a specific Hcy binding site.

Response: *We thank the reviewer for his valuable comment. To further verify the role of Arg¹⁶⁷ in the interaction between Hcy and AT1 receptor, new BRET assay was performed. BRET assay indicated that both Ang II and Hcy significantly induced β-arrestin 2 recruitment, whereas AT1 R167A mutation disrupted both Ang II- and Hcy-induced β-arrestin 2 recruitment (Figure 5F-G). Therefore, we agreed with reviewer's notion that Arg¹⁶⁷ is important for both Ang II and Hcy-activated AT1 signaling. Thus we have changed the sentence "specific binding site for Hcy" to "Hcy and AngII share a common binding site at Arg¹⁶⁷". Please refer to the line 19-20 on page 19 of revised manuscript.*

Although the Arg¹⁶⁷ is important for both the Hcy and AngII induced downstream signaling, it may interact with Hcy and AngII with different mode, based on our computational simulation results. To further distinguish the respective interaction of Arg¹⁶⁷ with Ang II and Hcy, the AT1 R167N and R167K mutations were applied in BRET assay. In our newly acquired experimental results, R167K mutation blocked both Ang II- and Hcy-induced β-arrestin 2 recruitment, whereas R167N mutation only abolished Hcy induced the β-arrestin 2 recruitment but not Ang II. These results indicated that the polar interaction of Arg¹⁶⁷ with the AngII is enough to support its downstream arrestin recruitment, whereas both polar and basic properties of Arg¹⁶⁷ is essential for its mediating Hcy induced arrestin signaling. These results are in consistent with our docking model that a salt bridge formed between Hcy and Arg¹⁶⁷. Taken together, these results disclosed that the Arg¹⁶⁷ is the key residue responsible for both Ang II and Hcy

interactions, but with distinct binding modes. Please refer to the text from line 20 of page 14 to line 12 of page 15 in the revised manuscript.

The results with the C289S mutation are more convincing. Do the authors have any data to suggest that homocysteine forms a disulfide with Cys289?

Response: *Many thanks for the reviewer's positive and suggestive comments. We have been trying several experiments to demonstrate the formation of disulfide bond between Hcy and Cys²⁸⁹.*

Firstly, we synthesized the peptide aa285-295 of AT1. Following incubation with 100 μ M Hcy in non-cell system, peptides were analyzed by MALDI-TOF mass spectrometry. Unfortunately, no S-homocysteinylation at Cys²⁸⁹ was measured. As the peptide lost its context in tertiary structure of the receptor, the S-homocysteinylation reaction might not happen due to the requirement of natural conformation of AT1 receptor for Hcy interaction.

Secondly, we overexpressed Flag-AT1 plasmid in HEK293A cell. After stimulation of Hcy (100 μ M) for 2 hours, we enriched AT1 receptor via immunoprecipitation with anti-Flag antibodies. Then we digested enriched protein fraction by trypsin and analyzed the peptides and amino acids information by MALDI-TOF mass spectrometry. However, the identified peptides covered only more than 20% of AT1 protein sequence, which did not include the peptide containing Cys289. Using mass spectrometry to determine the 7 transmembrane receptor is still a big technical challenge, normally required large amount protein purified by recombinant systems. We reasoned that the protein purified from cultured dish of HEK293 cells are not sufficient for mass spectrometry coverage.

Therefore, we have made the baculovirus system to produce recombinant AT1 receptor from insect cells. Once we have the recombinant proteins, we will perform both the mass spectrometry and crystallography studies to understand the interaction mode of Hcy with AT1 receptor. Whereas these experiments are still ongoing, we do not expect that we can have results in an immediate near future. These results may be shown in following published manuscripts.

Reviewer #2 (Remarks to the Author):

Hhcy has been previously shown to augment AAAs. The major conclusion of this manuscript is that Hhcy promotes AAA via direct stimulation of AT1a receptors. The reviewer acknowledges the structural studies demonstrate an ability of Hhcy to directly interact and stimulate this receptor. Also there is data to demonstrate that the augmented AAAs in Hhcy mice is inhibited by the absence of AT1a receptors. However, there is no direct evidence presented to demonstrate in vivo that Hhcy promotes AAA through direct AT1aR stimulation. It is surprising that this data has not been included.

Response: *Following reviewer's suggestion, a most direct method to prove the aggravated effect of HHcy on AAA formation via direct stimulation of AT1a is to apply AT1 receptor C289A and R167A double knock-in mice for evaluating whether HHcy-aggravated AAA formation would be inhibited in these knock-in mice. However, it is technically difficult to generate such double knock-in mice in short period. Alternatively, we performed new saturation binding assays with aortic membrane fraction of WT and AT1a^{-/-} mice with or without HHcy to confirm the direct interaction of Hcy and AT1a receptor in vivo. Consequence, HHcy attenuated the B_{max} value but increased the K_D value of [¹²⁵I]-Ang II in interaction with WT aortic membrane fraction (B_{max} of control vs. Hcy: 26268 ± 697 vs. 22315 ± 618 fmol-mg⁻¹ protein *P<0.05, non-specific binding was subtracted; K_D of control vs. Hcy: 36.4 ± 4.6 vs. 43.2 ± 3.9 nM, Figure 3C), whereas AT1a knock out decreased both B_{max} and K_D values as well as disrupted the effect of HHcy on B_{max} and K_D values of [¹²⁵I]-Ang II binding with aortic membrane fractions. The HHcy-induced B_{max} attenuation at the saturated concentration of Ang II suggested that Hcy directly interact with the aortic AT1a receptor in vivo and in physiological receptor concentrations. Together with our data that HHcy aggravated AAA was inhibited in AT1^{-/-} mice or application of AT1 blocker (Figure 1 and Online Figure 2-3), our results suggested that HHcy promotes AAA through direct AT1aR stimulation. Please refer to revised Figure 3C and paragraph 2 on page 12 of revise manuscript.*

Comments

1. The major conclusion of this manuscript is that AT1a receptors are stimulated independent of angiotensin II to promote AAA. For this statement to be included, there must be some study in which AAA are formed in mice that have had a manipulation to

reduce the production of angiotensin II. For example, administration with an ACE inhibitor.

Response: Following with reviewer's helpful suggestion, we applied ACEI enalapril to exclude the role of Ang II production involved in HHcy-aggravated AAA formation in vivo. We compared HHcy-exacerbated AAA formation with and without administration of enalapril in both elastase- and CaPO₄-induced aneurysmal models (Online Figure 12A). As expected, enalapril administration significantly decreased the blood pressure in both control and HHcy mice (Online Table 6). In contrast, administration of enalapril displayed no effect on HHcy-enhanced maximal aortic diameter enlargement in both aneurysmal models (Online Figure 12B-E), indicating that HHcy-aggravated AAA formation is independent on Ang II production. Please refer to line 11-18 on page 10 of revised manuscript.

2. In figure 1, data represented in figure 1B is that maximal aortic diameters were approximately 1 mm. Most measurements in this region for normal aorta would be in the 0.4 to 0.5 mm range. would be helpful to provide the starting size of each of these aortas. Based on these measurements, the infra renal aorta of all 4 groups is greatly expanded, with augmentation in the Hhcy group that are wild type fo At1a receptors. However, Figure 1C and D seems to indicate there are minimal elastin breaks in 3 groups. How can this be reconciled?

Response: Thanks the reviewer for this helpful suggestion. We measured the external diameters of infrarenal aorta in male mice aged at 8 weeks and 12 weeks without elastase induction and Hcy drinking. Consistent to previous reports (Circ Res 2016;119(10):1076-1088; Arterioscler Thromb Vasc Biol 2016;36(1):69-77), the diameters were 0.82 ± 0.03 mm and 0.89 ± 0.04 mm respectively. To verify the enlargement of infrarenal aorta of mice with AAA induction, we included the related data of WT and AT1a^{-/-} mice with sham surgery in revised Figure 1 and Online Table 1. As expected, elastase induced the expansion of abdominal aorta (WT vs. WT + elastase: 0.89 ± 0.04 [n=5] vs. 1.23 ± 0.04 [n=12] mm, *P<0.05) and the degradation of aortic elastin in WT mice whereas elastase-induced abdominal aorta enlargement was inhibited by AT1a knockout (AT1a^{-/-} vs. AT1a^{-/-} + elastase: 0.88 ± 0.05 [n=5] vs. 0.99 ± 0.05 [n=12], no significance). Please refer to the line 17-21 on page 6 of revised

manuscript.

Additionally, we presented more representative VVG images to show the elastin degradation in revised Figure 1C-D. The elastin staining of infrarenal aortas from WT and AT1a^{-/-} mice with sham surgery were included as well.

3. The authors cited a reference (25) that suggests potential effects of AT1aR in the aortic wall contributes to AAA to support their hypothesis. However, two groups have demonstrated that deletion of AT1a receptors in smooth muscle cells has minimal effects on aortic pathology in angiotensin II infused mice. Therefore, there should be additional information demonstrating that this receptor type in smooth muscle cells is relevant to the disease process.

Response: As reviewer suggested, vascular-resident but not bone marrow-derived AT1a receptors mediates AAA formation (*Arterioscler Thromb Vasc Biol* 2017;27(2):380-386), but recent data from cell-specific AT1a knockout mice suggest that AT1a deficiency in vascular endothelial cells, VSMCs or adventitial fibroblasts respectively has no effect on AngII-induced aortic pathologies, such as AAA, atherosclerosis and medial hyperplasia of descending aortas (*PLoS One* 2012;7(12):e51483; *Arterioscler Thromb Vasc Biol* 2015;35(9): 1995-2002). These findings indicate that a potential synergistic effect between different vascular cell types during overactivation of AT1 receptor signaling exists in aneurysmal disease states. In current study, we proposed the role of Hcy-induced AT1a receptor activation in whole vascular wall but not specific cell types, such as VSMCs. Please refer to paragraph 2 on page 24 of revised manuscript.

In the studies represented in online figure 4, aortic rings were incubated with Hhcy and effects were determined. For this to be meaningful, there has to be additional data demonstrating similar changes during incubation with angiotensin II.

Response: Following reviewer's helpful suggestion, we included the data of MCP-1 and IL-6 production of aortic rings stimulated by Ang II (1 μ M). Consequently, Ang II and Hcy exhibited similar effects on MCP-1 and IL-6 induction. Please refer to the revised Online Figure 4.

4 Telmisartan was used in most experiments to block AT1R, with a single exception that candesartan was used without providing rationale why either was used. More importantly, it is well-known that telmisartan has strong PPAR γ activation effect. Therefore, data interpretation should at least discuss the potential off-target effects.

Response: *We agree with reviewer's concern for this important point. To evaluate whether other sartans had the similar functions of telmisartan on inhibition of Hcy-induced AT1 activation, candesartan and losartan were applied. As a result, both sartans markedly blocked Hcy-induced PKC and ERK1/2 phosphorylation and NFAT activation similar to telmisartan (Online Figure 10C-D). Thus, these sartans seems have the similar effects on Hcy-induced AT1 activation.*

As reviewer suggested that telmisartan has strong PPAR γ activation effect (Cardiovasc Res 2011;90(1):122-129), we applied PPAR γ agonist rosiglitazone (RSG) to examine whether it can inhibit the activation of AT1 receptor by Hcy. Distinct with telmisartan, RSG did not inhibit Hcy-induced PKC and ERK1/2-MAPK phosphorylation and NFAT activation, implying that the inhibitive effect of telmisartan is independent on its side-effect on activating PPAR γ (Online Figure 10A-B). Please refer to the text from line 18 on page 9 to line 3 on page 10 of revised manuscript.

5. Online Figure 5: The authors performed ex vivo experiment and stated that AngII secretion from aortic ring explants was not altered by homocysteine within 12 hours, which does not rule out increase of AngII after 12 hours. Additionally, the authors measured mRNA of AGT, ACE, and AT1aR, but not renin, in the aortic explant. If renin is not present in the aortic ring, how was AngII produced?

Response: *Following reviewer's suggestion, we additionally measured renin gene expression by qPCR and determined the Ang II secretion of infrarenal aorta with Hcy treatment for 24 and 48 hours ex vivo. Similar to AT1a, AGT and ACE, renin did not alter at mRNA level within 12 hours, whereas it was up-regulated after a longer stimulation time (24 and 48 hours) (Online Figure 5D). Accordingly, Ang II secretion of aortic ring explants was not altered by Hcy within 24 hours but significantly elevated under 48-hour treatment of Hcy (Online Figure 5E).*

6. No immunofluorescent staining method is described. The specificity of all antibodies

needs to be validated.

Response: We apologize for this unclearly description. The immunofluorescent images in revised Figure 2H-I were captured under microscopy via the overexpression of human AT1 fused with mCherry, β -arrestin 2 fused with GFP and mouse AT1a fused with GFP plasmids, rather than application of antibodies. We have clarified it in the legends of Figure 2H-I.

7. There is considerable skepticism about the accuracy of measuring angiotensin II. The authors have used this using a kit in which the reviewer can find no information. It is important to have a full description of this assay. For these measurements to be generally accepted, it would be helpful to provide validation studies.

Response: We applied a commercial radioimmunoassay (RIA) kit (Bühlmann Laboratories, Basel, Switzerland) to measure the concentrations of Ang II. We described the detailed procedure of the measurement process in “Analysis of Ang II Concentration” part of the revised Methods. The samples, calibrators and controls were first preincubated for 16 hours with an anti-Ang II antibody. [¹²⁵I]-Ang II was then added and competes with Ang II present in samples, calibrators and controls for the same antibody binding sites in a second 6 hours incubation step. After this incubation, a solid-phase secondary antibody was added to the mixture. The antibody-bound fraction was precipitated and counted in a gamma-counter. The radioactive values in calibrators were regressed with respective Ang II concentrations into the standard curve. Based on it, the concentrations of Ang II were calculated. The quality of current commercial kit has been validated in previous study (Horm Res. 1991;36(1-2):78-9).

8. It should be stated where the aortic rings are derived from. If these rings are derived from regions that are aneurysm-resistant, the relevance of findings in these tissues must be discussed.

Response: Actually we isolated aortic rings from abdominal aorta where AAA usually happens. We added the related information in figure legends and methods accordingly. Please refer to revised methods of Gelatin Zymography and figure legend of Fig 1E-H, Online Figure 4 and Online Figure 5.

9. On page 30 the last paragraph, the authors state that “Therefore, in addition to lowering total plasma Hcy, these patients may benefit from using AT1 receptor blockers

rather than ACEIs, as our data showed that enalapril does not inhibit Hcy-induced AT1 receptor activation, while telmisartan does.” There has no data to support this conclusion.

Response: *To further verify the role of ACEI in HHcy-aggravated AAA formation, we compared HHcy-exacerbated AAA formation with and without administration of enalapril in both elastase- and CaPO4-induced aneurysmal models (Online Table 6 and Online Figure 12A). As a consequence, administration of enalapril displayed no effect on HHcy-enhanced the enlarged maximal aortic diameter in both aneurysmal models (Online Figure 12B-E), indicating that HHcy-aggravated AAA formation is independent on Ang II production. These results suggested that HHcy patients might not benefit from ACEIs for AAA. Please refer to line 11-18 on page 10 of revised manuscript.*

Reviewer #3 (Remarks to the Author):

This is a review of the computational portions of the paper. The authors used small molecule docking with AutoDock 4.2 to identify potential homocysteine binding sites on AT1 using both global docking and local refinement. AT1 is a membrane-embedded GPCR, so docking to lipid-facing regions is unusual, and any predicted conformations in the lipid bilayer region would be suspect. Fortunately, only cluster 1 looks to suffer from this problem. The most interesting cluster (cluster 3) was in the solvent-facing surface of AT1, so this is a reasonable prediction. The follow-up experiments support the binding location and the identification of Arg167 as a key binding residue. Additionally, MD simulations of the HCY at cluster 3 were stable.

Response: *Thanks for reviewer's positive comments.*

The authors also suggest that HCY may make a disulfide bond with C289. In Fig 5J, these atoms are not close enough to bond, but it is conceivable that the HCY position would be dynamic enough to allow bonding.

Response: *Thank you for your positive comments and helpful suggestions. We have included corresponding depiction in our main manuscript in line 2-11 on Page 18 of revised manuscript.*

Overall, this paper uses fairly standard computational methods in reasonable manners to support their stated conclusions.

Response: *Thanks for reviewer's positive comments.*

One minor edit on p. 23 line 22: "stimulation" should be "simulation"

Response: *Thank for reviewer's valuable suggestion, we have changed the word accordingly. Please refer to last paragraph on page 17 and paragraph 1 on page 33 of revised manuscript.*

Reviewers' comments:

Reviewer #1 (Remarks to the Author):

The authors have responded comprehensively to the prior critique and the manuscript has been improved.

Although there is convincing evidence for interaction (binding) of Hcy to both Arg167 and Cys289, there is still no direct evidence that Hcy forms a disulfide bond with Cys289. Therefore, the Abstract should be revised to indicate that the formation of a disulfide is postulated, not "revealed" or "proven."

In the Discussion (page 21), the section discussing the large difference in the binding affinity of Hcy and Ang II to AT1 (5 logs) should be expanded to acknowledge several limitations. First, if, as the authors suggest in their response to Reviewer 1, the active metabolite of Hcy is the free thiol form of homocysteine, then the plasma concentration is not 5 logs higher than Ang II (maybe 3 logs at best). Second, the authors should add a discussion about the relative concentrations of total Hcy and free thiol homocysteine and acknowledge that we still do not know which form(s) of Hcy are active with respect to AT1.

Reviewer #2 (Remarks to the Author):

The authors have responded extensively. The major conclusion of the manuscript is that homocysteine stimulates AT1a receptor to promote AAA. This would be consistent with the elegant in vitro studies. However, the extrapolation of this to the in vivo setting has confounders. To overcome this concern, a study was added in which enalapril was administered and found to have no effect on AAAs. The interpretation of this negative study relies on knowing that the dose of enalapril, in the mode administered, causes a persistent and profound reduction in AngII production. Currently, the only evidence is that there are small blood pressure reduction (using a tail cuff technique) at an undefined time after enalapril administration. This potentially important study would have much greater credibility if the negative enalapril study would have definitive evidence that it persistently inhibited ACE.

Comments:

1. The added study administered enalapril by a daily gavage. The half life of enalapril is approximately 11 hours, so this mode of administration may lead to large "peak and trough" plasma concentrations over the day. In contrast, telmisartan was administered in drinking water, which is more likely to give persistent drug level in plasma over the day. Why were these two different routes used for the two drugs? Can evidence be provided that AngII production was suppressed extensively and constantly? It would help to provide information on the timing of the measurement of the small reductions in blood pressure relative to the drug administration. The measurement of blood pressure in mice being gavaged each day has potential confounders. ACE inhibition usually promotes large increases in plasma renin activity. Can this be measured? Overall, the impact of this manuscript would be greatly enhanced by the demonstration of profound and persistent inhibition of AngII production.

2. Why was enalapril 5 mg/kg/d used? It is unclear whether enalapril 5 mg/kg/d reached its maximal effects on suppressing AngII production, although the reviewer acknowledges that tail-cuff blood pressure measurements showed a modest reduction of systolic blood pressure.

3. No information was provided for tail-cuff blood pressure measurements. The accuracy and reliability of this technique require standardization of protocol. Please refer to Kurtz et al AHA Statement for Blood Pressure Measurement in Animal published in AHA journals in 2005.

Reviewer #3 (Remarks to the Author):

The authors have addressed all concerns.

Point-by-point Response

Reviewers' comments:

Reviewer #1 (Remarks to the Author):

The authors have responded comprehensively to the prior critique and the manuscript has been improved.

Response: Many thanks for the reviewer's positive comments.

Although there is convincing evidence for interaction (binding) of Hcy to both Arg167 and Cys289, there is still no direct evidence that Hcy forms a disulfide bond with Cys289. Therefore, the Abstract should be revised to indicate that the formation of a disulfide is postulated, not "revealed" or "proven."

Response: Following the reviewer's suggestion, we changed 'revealed' into 'suggested'. Please refer to the line 12 of revised Abstract on Page 3.

In the Discussion (page 21), the section discussing the large difference in the binding affinity of Hcy and Ang II to AT1 (5 logs) should be expanded to acknowledge several limitations. First, if, as the authors suggest in their response to Reviewer 1, the active metabolite of Hcy is the free thiol form of homocysteine, then the plasma concentration is not 5 logs higher than Ang II (maybe 3 logs at best). Second, the authors should add a discussion about the relative concentrations of total Hcy and free thiol homocysteine and acknowledge that we still do not know which form(s) of Hcy are active with respect to AT1.

Response: We completely agreed with the reviewer's suggestion. According to the literatures (J Biol Chem 277: 30425–30428; J Physiol Pharmacol 59 (Suppl 9): 155–

167), plasma Hcy exists in three different forms defined as free thiol Hcy (1%), protein-bound Hcy (80-90%) and oxidized form of Hcy (10-20%). The plasma concentration of total Hcy is 5 logs higher than Ang II, whereas the free thiol Hcy in plasma may be only 10^3 times higher than Ang II. We acknowledged that currently we still have not known which form of Hcy contributes to the interaction with AT1 receptor in vivo. Please refer to the line 20 of Page 21 to the line 4 of Page 22 in revised Discussion.

Reviewer #2 (Remarks to the Author):

The authors have responded extensively. The major conclusion of the manuscript is that homocysteine stimulates AT1a receptor to promote AAA. This would be consistent with the elegant in vitro studies. However, the extrapolation of this to the in vivo setting has cofounders. To overcome this concern, a study was added in which enalapril was administered and found to have no effect on AAAs. The interpretation of this negative study relies on knowing that the dose of enalapril, in the mode administered, causes a persistent and profound reduction in AngII production. Currently, the only evidence is that there are small blood pressure reduction (using a tail cuff technique) at an undefined time after enalapril administration. This potentially important study would have much greater credibility if the negative enalapril study would have definitive evidence that it persistently inhibited ACE.

Response: We thank the reviewer for the valuable suggestions, and we further have performed additional experiments accordingly.

Comments

1. The added study administered enalapril by a daily gavage. The half life of enalapril is approximately 11 hours, so this mode of administration may lead to large “peak and trough” plasma concentrations over the day. In contrast, telmisartan was administered in drinking water, which is more likely to give persistent drug level in plasma over the day. Why were these two different routes used for the two drugs? Can evidence be provided that AngII production was suppressed extensively and constantly? It would help to provide information on the timing of the measurement of the small reductions in blood pressure relative to the drug administration. The measurement of blood pressure in mice being gavaged each day has potential cofounders. ACE inhibition usually promotes large increases in plasma renin activity. Can this be measured? Overall, the impact of this manuscript would be greatly enhanced by the demonstration of profound

and persistent inhibition of AngII production.

Response: *We totally agreed with reviewer that daily gavage of enalapril might lead to a nonpersistent effect on inhibiting Ang II production, although we found the decrease of systolic blood pressure following gavage treatment. As suggested by reviewer, to achieve more extensive and constant effects, we dissolved enalapril in drinking water (0.15 g/L) and gave an estimated daily dose of 30 mg/kg/d to mice according to previous studies (Science. 2011;332(6027):361-5; Am J Physiol Endocrinol Metab 2012;302:E500–E509; Clinical Science 2003;104:109–118). We additionally measured Ang II production and renin activity in the plasma from mice administrated with 30 mg/kg/d enalapril in drink water or daily gavage of 5 mg/kg/d enalapril respectively. As expected, enalapril at 30 mg/kg/d displayed more profound inhibition on Ang II production and enhance of renin activity compared to gavage of 5 mg/kg/d dose (Response Figure). However, the administration of 30 mg/kg/d enalapril in drinking water still displayed no effect on HHcy-aggravated infrarenal abdominal aortic dilation in elastase-induced model, indicating that HHcy-aggravated AAA formation is independent on Ang II production (Online Figure 12). We replaced previous enalapril results with current data of administration of 30 mg/kg/d enalapril in drinking water in Online Figure 12. Please also refer to revised Results in line 11-20 on Page 10, Online Table 6 and revised Method information on Page 27 and 30.*

2. Why was enalapril 5 mg/kg/d used? It is unclear whether enalapril 5 mg/kg/d reached its maximal effects on suppressing AngII production, although the reviewer acknowledges that tail-cuff blood pressure measurements showed a modest reduction of systolic blood pressure.

Response: *Currently the dose of enalapril applied in mice in vivo varies between 5-30 mg/kg/d in different studies. The dosage of enalapril at 5 mg/kg/d was used following the previous reports (Am J Physiol Lung Cell Mol Physiol 2002;282:L1209–L1221;*

Clinical Science 2003;104:109–118). Indeed, enalapril at this dosage could decrease the angiotensin II level and blood pressure to some degree. Nevertheless, as demonstrated by previous studies (Clinical Science 2003;104:109–118; Science. 2011;332(6027):361-5.), enalapril at 5 mg/kg/d does not reach its maximal effects on suppressing AngII production. Following previous findings (Clinical Science 2003;104:109–118; Science. 2011;332(6027):361-5.), we instead dissolved enalapril in drinking water (0.15g/L) as an estimated daily dose of 30 mg/kg/d to mice calculated based on the body weight of 25 g and daily intake volume of water as 5 ml. The dosage of 30 mg/kg/d almost maximally inhibited angiotensin I-induced the elevation of blood pressure in mice. We found enalapril at 30 mg/kg/d further inhibited Ang II production and increased plasma renin activity as compared to gavage of 5 mg/kg/d dose (Response Figure). However, the administration of 30 mg/kg/d enalapril in drinking water still displayed no effect on HHcy-aggravated aortic enlargement in elastase-induced model (Online Figure 12). Please also refer to revised Results in line 11-20 on Page 10, Online Table 6 and revised Method information on Page 27 and 30.

3. No information was provided for tail-cuff blood pressure measurements. The accuracy and reliability of this technique require standardization of protocol. Please refer to Kurtz et al AHA Statement for Blood Pressure Measurement in Animal published in AHA journals in 2005.

Response: *Following reviewer's suggestion, we expanded and clarified the methodology of tail-cuff measurement of systolic blood pressure in more detail in Method part on Page 27. Especially, we have mainly paid attention to the following points. ① Our mice were exposed to the measurement procedures everyday for consecutive 7 days prior to the beginning of an experiment. ② The measurement was performed using clean equipment free from foreign scent and blood odor. ③ The measurement was performed by same person at same time each day. ④ The final blood pressure was average number of 15 measurements in a 3-day recording session.*

Moreover, we refer to the AHA Statement (*Arterioscler Thromb Vasc Biol.* 2005;25:e22-e33) suggested by reviewer, and our procedure is actually consistent with the statement.

Response Figure

Reviewer #3 (Remarks to the Author):

The authors have addressed all concerns.

Response: We appreciated the reviewer's positive comment.

REVIEWERS' COMMENTS:

Reviewer #2 (Remarks to the Author):

No further comments.